# A Study of Biologically Plausible Neural Network: The Role and Interactions of Brain-Inspired Mechanisms in Continual Learning

**Fahad Sarfraz**[‡1,2]**, Elahe Arani**[‡1,2]**, Bahram Zonooz**[1,2]
*fahad.sarfraz@navinfo.eu, {e.arani, bahram.zonooz}@gmail.com*
[1] *Advanced Research Lab, NavInfo Europe, Eindhoven, The Netherlands*
[2] *Department of Mathematics and Computer Science, Eindhoven University of Technology, The Netherlands*
[‡] *Contributed equally.*

**Reviewed on OpenReview:** *https://openreview.net/forum?id=DJr6zorJM2*

## Abstract

Humans excel at continually acquiring, consolidating, and retaining information from an ever-changing environment, whereas artificial neural networks (ANNs) exhibit catastrophic forgetting. There are considerable differences in the complexity of synapses, the processing of information, and the learning mechanisms in biological neural networks and their artificial counterparts, which may explain the mismatch in performance. We consider a biologically plausible framework that constitutes separate populations of exclusively excitatory and inhibitory neurons that adhere to Dale's principle, and the excitatory pyramidal neurons are augmented with dendritic-like structures for context-dependent processing of stimuli. We then conduct a comprehensive study on the role and interactions of different mechanisms inspired by the brain, including sparse non-overlapping representations, Hebbian learning, synaptic consolidation, and replay of past activations that accompanied the learning event. Our study suggests that the employing of multiple complementary mechanisms in a biologically plausible architecture, similar to the brain, may be effective in enabling continual learning in ANNs. [1]

## 1 Introduction

The human brain excels at continually learning from a dynamically changing environment, whereas standard artificial neural networks (ANNs) are inherently designed for training from stationary i.i.d. data. Sequential learning of tasks in continual learning (CL) violates this strong assumption, resulting in catastrophic forgetting. Although ANNs are inspired by biological neurons (Fukushima, 1980), they omit numerous details of the design principles and learning mechanisms in the brain. These fundamental differences may account for the mismatch in performance and behavior.

Biological neural networks are characterized by considerably more complex synapses and dynamic context-dependent processing of information. In addition, individual neurons have a specific role. Each presynaptic neuron has an exclusive excitatory or inhibitory impact on its postsynaptic partners, as postulated by Dale's principle (Strata et al., 1999). Furthermore, distal dendritic segments in pyramidal neurons, which comprise the majority of excitatory cells in the neocortex, receive additional context information and enable context-dependent processing of information. This, in conjunction with inhibition, allows the network to learn task-specific patterns and avoid catastrophic forgetting (Yang et al., 2014; Iyer et al., 2022; Barron et al., 2017). Furthermore, replay of non-overlapping and sparse neural activities of previous experiences in the neocortex and hippocampus is considered to play a critical role in memory formation, consolidation, and retrieval (Walker & Stickgold, 2004; McClelland et al., 1995). To protect information from erasure,

---

[1] We will make the code available upon acceptance.

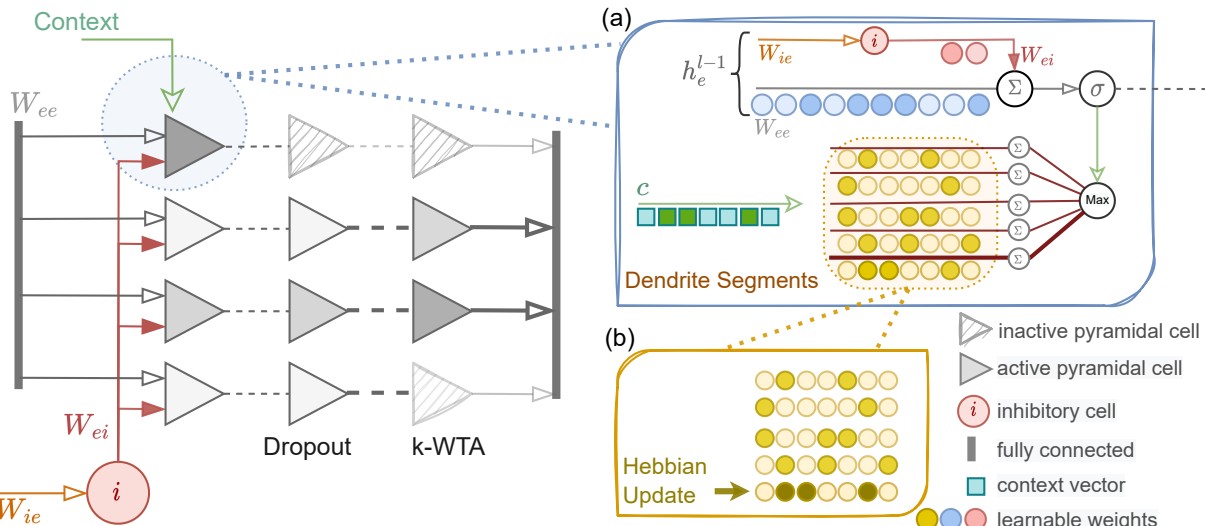

Figure 1: Architecture of one hidden layer in the biologically plausible framework. Each layer consists of separate populations of exclusively excitatory pyramidal cells and inhibitory neurons, which adhere to Dale's principle. The shade indicates the strength of weights or activations, with a darker shade indicating a higher value. (a) The pyramidal cells are augmented with dendritic segments, which receive an additional context signal $c$ and the dendritic segment whose weights are most aligned with the context vector (bottom row) is selected to modulate the output activity of the feedforward neurons for context-dependent processing of information. (b) The Hebbian update step further strengthens the association between the context and the winning dendritic segment with maximum absolute value (indicated with a darker shade in the bottom row). Finally, Heterogeneous dropout keeps the activation count of each pyramidal cell (indicated with the gray shade) and drops the neurons that were most active for the previous task (the darkest shade dropped) to enforce non-overlapping representations. The top-k remaining cells then project to the next layer (increased shade). This provides us with a more biologically plausible framework within which we can study the role of different brain-inspired mechanisms and provide insights for designing new CL methods.

the brain employs synaptic consolidation, in which plasticity rates are selectively reduced in proportion to strengthened synapses (Cichon & Gan, 2015).

Thus, we study the role and interactions of different mechanisms inspired by the brain in a biologically plausible framework in a CL setup. The underlying model constitutes separate populations of exclusively excitatory and inhibitory neurons in each layer, which adheres to Dale's principle (Cornford et al., 2020) and excitatory neurons (mimicking pyramidal cells) are augmented with dendrite-like structures for context-dependent processing of information (Iyer et al., 2022). Dendritic segments process an additional context signal encoding task information and subsequently modulate the feedforward activity of the excitatory neuron (Figure 1). We then systematically study the effect of controlling the overlap in representations, employing the "fire together, wire together" learning paradigm and employing experience replay and synaptic consolidation. Our study shows that:

i. An ANN architecture equipped with context-dependent processing of information by dendrites and adhering to Dale's principle can learn effectively in CL setup. Importantly, accounting for the discrepancy in the effect of weight changes in excitatory and inhibitory neurons further reduces forgetting in CL.

ii. Enforcing different levels of activation sparsity in the hidden layers using k-winner-take-all activations and employing a complementary dropout mechanism (Heterogeneous Dropout) that encourages the model to use a different set of active neurons for each task can effectively control the overlap in representations, and hence reduce interference while allowing for resusability.

iii. Task similarities need to be considered when enforcing such constraints to allow for a balance between forwarding transfer and interference.

iv. Mimicking the ubiquitous "fire together, wire together" learning rule in the brain through a Hebbian update step on the connection between context signal and dendritic segments, which further strengthens context gating and facilitates the formation of task-specific subnetworks.

v. We show that employing both synaptic consolidation with importance measures adjusted to take into account the discrepancy in the effect of weight changes and a replay mechanism in a context-specific manner is critical for consolidating information across different tasks, especially for challenging CL settings.

Our study suggests that employing multiple complementary mechanisms in a biologically plausible architecture, similar to what is believed to exist in the brain, can be effective in enabling CL in ANNs. To the best of our knowledge, we are the first to provide a comprehensive study of the integration of different brain-inspired mechanisms in a biologically plausible architecture in a CL setting.

## 2 Biologically Plausible Framework for CL

We provide details of the biologically plausible framework within which we conduct our study.

### 2.1 Dale's Principle

Biological neural networks differ from their artificial counterparts in the complexity of synapses and the role of individual units. In particular, the majority of neurons in the brain adhere to Dale's principle, which posits that presynaptic neurons can only have an exclusive excitatory or inhibitory impact on their postsynaptic partners (Strata et al., 1999). Several studies show that the balanced dynamics (Murphy & Miller, 2009; Van Vreeswijk & Sompolinsky, 1996) of excitatory and inhibitory populations provide functional advantages, including efficient predictive coding (Boerlin et al., 2013) and pattern learning (Ingrosso & Abbott, 2019). Furthermore, inhibition is hypothesized to play a role in alleviating catastrophic forgetting (Barron et al., 2017). Standard ANNs, however, lack adherence to Dale's principle, as neurons contain both positive and negative output weights, and signs can change while learning.

Cornford et al. (2020) incorporate Dale's principle into ANNs (referred to as DANNs), which take into account the distinct connectivity patterns of excitatory and inhibitory neurons (Tremblay et al., 2016) and perform comparable to standard ANNs in the benchmark object recognition task. Each layer $l$ comprises of a separate population of excitatory, $h_e^l \in \mathbb{R}_+^{n_e}$, and inhibitory $h_i^l \in \mathbb{R}_+^{n_i}$ neurons, where $n_e \gg n_i$ and synaptic weights are strictly non-negative. Similar to biological networks, while both populations receive excitatory projections from the previous layer ($h_e^{l-1}$), only excitatory neurons project between layers, whereas inhibitory neurons inhibit the activity of excitatory units of the same layer. Cornford et al. (2020) characterized these properties by three sets of strictly positive weights: excitatory connections between layers $W_{ee}^l \in \mathbb{R}_+^{n_e \times n_e}$, excitatory projection to inhibitory units $W_{ie}^l \in \mathbb{R}_+^{n_i \times n_e}$, and inhibitory projections within the layer $W_{ei}^l \in \mathbb{R}_+^{n_e \times n_i}$. The output of the excitatory units is impacted by the subtractive inhibition from the inhibitory units:

$$z^l = (W_{ee}^l - W_{ei}^l W_{ie}^l)h_e^{l-1} + b^l \tag{1}$$

where $b^l \in \mathbb{R}^{n_e}$ is the bias term. Figure 1 shows the interactions and connectivity between excitatory pyramidal cells (triangle symbol) and inhibitory neurons (denoted by $i$).

We aim to employ DANNs as feedforward neurons to show that they can also learn in a challenging CL setting and performance comparable to standard ANNs and provide a biologically plausible framework for further studying the role of inhibition in alleviating catastrophic forgetting.

### 2.2 Active Dendrites

The brain employs specific structures and mechanisms for context-dependent processing and routing of information. The prefrontal cortex, which plays an important role in cognitive control (Miller & Cohen,

2001), receives sensory inputs as well as contextual information, which allows it to choose the most relevant sensory features for the present task to guide actions (Mante et al., 2013; Fuster, 2015; Siegel et al., 2015; Zeng et al., 2019). Of particular interest are pyramidal cells, which represent the most populous members of the excitatory family of neurons in the brain (Bekkers, 2011). The dendritic spines in pyramid cells exhibit highly non-linear integrative properties that are considered important for learning task-specific patterns (Yang et al., 2014). Pyramidal cells integrate a range of diverse inputs into multiple independent dendritic segments, allowing contextual inputs in active dendrites to modulate the response of a neuron, making it more likely to fire. However, standard ANNs are based on a point neuron model (Lapique, 1907) which is an oversimplified model of biological computations and lacks the sophisticated non-linear and context-dependent behavior of pyramidal cells.

Iyer et al. (2022) model these integrative properties of dendrites by augmenting each neuron with a set of dendritic segments. Multiple dendritic segments receive additional contextual information, which is processed using a separate set of weights. The resultant dendritic output modulates the feedforward activation, which is computed by a linearly weighted sum of the feedforward inputs. This computation results in a neuron where the magnitude of the response to a given stimulus is highly context-dependent. To enable task-specific processing of information, the prototype vector for task $\tau$ is evaluated by taking the element-wise mean of the tasks samples, $\mathcal{D}_\tau$ at the beginning of the task and then subsequently provided as context during training.

$$c_\tau = \frac{1}{|\mathcal{D}_\tau|} \sum_{x \in \mathcal{D}_\tau} x \tag{2}$$

During inference, the closest prototype vector to each test sample, $x'$, is selected as the context using the Euclidean distance among all task prototypes, $\boldsymbol{C}$, stored in memory.

$$c' = \arg\min_{c_\tau} \|\boldsymbol{x}' - \boldsymbol{C}_\tau\|_2 \tag{3}$$

Following Iyer et al. (2022), we augment the excitatory units in each layer with dendritic segments (Figure 1 (a)). The feedforward activity of excitatory units is modulated by dendritic segments, which receive a context vector. Given the context vector, each dendritic segment $j$ computes $u_j^T c$, given weight $u_j \in \mathbb{R}^d$ and the context vector $c \in \mathbb{R}^d$ where $d$ is the dimensions of the input image. For excitatory neurons, the dendritic segment with the highest response to the context (maximum absolute value with the sign retained) is selected to modulate output activity.

$$h_e^l = k\text{-}WTA(z_l \times \sigma(u_\kappa^T c)), \qquad \text{where } \kappa = \arg\max_j |u_j^T c| \tag{4}$$

where $\sigma$ is the sigmoid function (Han & Moraga, 1995) and $k\text{-}WTA(.)$ is the k-Winner-Take-All activation function (Ahmad & Scheinkman, 2019) which propagates only the top $k$ neurons and sets the rest at zero. This provides us with a biologically plausible framework where, similar to biological networks, the feedforward neurons adhere to Dale's principle, and the excitatory neurons mimic the integrative properties of active dendrties for context-dependent processing of stimuli.

## 3   Continual Learning Settings

To study the role of different components inspired by the brain in a biologically plausible NN for CL and gauge their roles in the performance and characteristics of the model, we conduct all our experiments under uniform settings. Implementation details and experimental setup are provided in Appendix. We evaluate the models on two CL scenarios. **Domain incremental learning (Domain-IL)** refers to the CL scenario in which the classes remain the same in subsequent tasks but the input distribution changes. We consider Rot-MNIST which involves classifying the 10 digits in each task with each digit rotated by an angle between 0 and 180 degrees, and Perm-MNIST which applies a fixed random permutation to the pixels for each task. Importantly, there are different variants of Rot-MNIST with varying difficulties. We incrementally rotate the digits to a fixed degree, i.e. {0, 8, ..., (N-1)*8} for task $\{\tau_1, \tau_2, .., \tau_N\}$ which is substantially more challenging than random sampling rotations. Importantly, the Rot-MNIST dataset captures the notion of

Table 1: Effect of each component of the biologically plausible framework on different datasets with varying number of tasks. We first show the effect of utilizing feedforward neurons adhering to Dale's principle in conjunction with *Active Dendrites* to form the framework within which we evaluate the individual effect of brain-inspired mechanisms (Hebbian Update, Heterogeneous Dropout (HD), Synaptic Consolidation (SC), Experience Replay (ER) and Consistency Regularization (CR)) before combining them all together to forge Bio-ANN. For all experiments, we set the percentage of active neurons at 5. We provide the average task performance and 1 std of five runs. We also demonstrate performance gains with the brain-inspired mechanisms on top of standard ANNs in Table 6 in Appendix

| Method | Rot-MNIST | | | Perm-MNIST | | | Seq-MNIST |
|---|---|---|---|---|---|---|---|
| | 5 Tasks | 10 Tasks | 20 Tasks | 5 Tasks | 10 Tasks | 20 tasks | |
| Joint | $98.27_{\pm 0.07}$ | $98.25_{\pm 0.06}$ | $98.17_{\pm 0.11}$ | $97.64_{\pm 0.19}$ | $97.61_{\pm 0.15}$ | $97.59_{\pm 0.08}$ | $94.53_{\pm 0.50}$ |
| SGD | $93.79_{\pm 0.32}$ | $74.13_{\pm 0.36}$ | $51.89_{\pm 0.36}$ | $77.96_{\pm 8.34}$ | $76.42_{\pm 2.54}$ | $65.18_{\pm 2.12}$ | $19.83_{\pm 0.04}$ |
| Active Dendrites | $92.58_{\pm 0.26}$ | $71.06_{\pm 0.52}$ | $48.18_{\pm 0.52}$ | $95.71_{\pm 0.27}$ | $94.41_{\pm 0.21}$ | $91.74_{\pm 0.34}$ | $19.97_{\pm 0.29}$ |
| + Dale's Principle | $92.28_{\pm 0.27}$ | $70.78_{\pm 0.23}$ | $48.79_{\pm 0.27}$ | $96.18_{\pm 0.14}$ | $95.20_{\pm 0.11}$ | $92.44_{\pm 0.20}$ | $19.79_{\pm 0.08}$ |
| + Hebbian Update | $92.58_{\pm 0.35}$ | $71.22_{\pm 0.86}$ | $48.88_{\pm 0.90}$ | $95.90_{\pm 0.24}$ | $94.72_{\pm 0.29}$ | $92.55_{\pm 0.47}$ | $19.88_{\pm 0.02}$ |
| + HD | $93.24_{\pm 0.25}$ | $75.50_{\pm 0.74}$ | $51.11_{\pm 0.76}$ | $96.58_{\pm 0.17}$ | $95.94_{\pm 0.24}$ | $93.20_{\pm 0.32}$ | $38.45_{\pm 2.71}$ |
| + SC | $93.34_{\pm 0.57}$ | $75.94_{\pm 1.15}$ | $64.99_{\pm 2.19}$ | $96.54_{\pm 0.29}$ | $96.14_{\pm 0.47}$ | $95.43_{\pm 0.49}$ | $27.31_{\pm 2.20}$ |
| + ER | $95.21_{\pm 0.28}$ | $90.99_{\pm 0.51}$ | $83.45_{\pm 0.44}$ | $96.72_{\pm 0.13}$ | $96.04_{\pm 0.17}$ | $94.26_{\pm 0.78}$ | $86.93_{\pm 0.82}$ |
| + ER + CR | $96.48_{\pm 0.55}$ | $93.87_{\pm 0.25}$ | $89.39_{\pm 0.23}$ | $97.23_{\pm 0.30}$ | $96.93_{\pm 0.31}$ | $96.13_{\pm 0.05}$ | $89.60_{\pm 0.73}$ |
| Bio-ANN | $\mathbf{96.82}_{\pm 0.14}$ | $\mathbf{94.64}_{\pm 0.23}$ | $\mathbf{91.32}_{\pm 0.26}$ | $\mathbf{97.33}_{\pm 0.04}$ | $\mathbf{97.07}_{\pm 0.05}$ | $\mathbf{96.51}_{\pm 0.03}$ | $\mathbf{89.90}_{\pm 0.24}$ |

similarity in subsequent tasks, where the similarity between two tasks is defined by the difference in their degree of rotation, whereas each task in Perm-MNIST is independent. We also consider the challenging **Class incremental learning (Class-IL)** scenario where new classes are added with each subsequent task and the agent must learn to distinguish not only amongst the classes within the current task but also across all learned tasks. Seq-MNIST divides the MNIST classification into 5 tasks with 2 classes for each task.

## 4 Empirical Evaluation

To investigate the impact of the different components inspired by the brain, we use the aforementioned biologically plausible framework and study the effect on the performance and characteristics of the model.

### 4.1 Effect of Inhibitory Neurons

We first study whether feedforward networks with separate populations of excitatory and inhibitory units can work well in the CL setting. Importantly, we note that when learning a sequence of tasks with inhibitory neurons, it is beneficial to take into account the disparities in the degree to which updates to different parameters affect the layer's output distribution (Cornford et al., 2020) and hence forgetting. Specifically, since $W_{ie}^l$ and $W_{ei}^l$ affect the output distribution to a higher degree than $W_{ee}^l$, we reduce the learning rate for these weights after the first task (see Appendix).

Table 1 shows that models with feedforward neurons adhering to Dale's principle perform on par with standard neurons and can also further mitigate forgetting in conjunction with Active Dendrites when the quality of context signal is high (in case of Permuted-MNIST). Note that this gain comes with considerably fewer parameters and context-dependent processing, as we keep the number of neurons in each layer the same, and only excitatory neurons ($\sim$90%) are augmented with dendritic segments. For 20 tasks, Active Dendrite with Dale's principle reduces the parameters from $\sim$70M to less than $\sim$64M parameters. We hypothesize that having separate populations within a layer enables them to play a specialized role. In particular, inhibitory neurons can selectively inhibit certain excitatory neurons based on stimulus, which can further facilitate the formation of task-specific subnetworks and complement the context-dependent processing of information by dendritic segments.

Table 2: Effect of different levels of sparsity in activations on the performance of the model. Columns show the ratio of active neurons ($k$ in k-WTA activation), and rows provide the number of tasks.

| $k$ #Tasks | Rot-MNIST | | | | Perm-MNIST | | | |
|---|---|---|---|---|---|---|---|---|
| | 0.05 | 0.10 | 0.20 | 0.50 | 0.05 | 0.10 | 0.20 | 0.50 |
| 5 | $92.28_{\pm 0.27}$ | $92.26_{\pm 0.31}$ | $\mathbf{92.79}_{\pm 0.44}$ | $92.26_{\pm 0.65}$ | $95.77_{\pm 0.33}$ | $\mathbf{96.32}_{\pm 0.20}$ | $90.29_{\pm 6.07}$ | $74.51_{\pm 13.55}$ |
| 10 | $70.78_{\pm 0.23}$ | $71.95_{\pm 1.54}$ | $\mathbf{73.32}_{\pm 0.69}$ | $71.61_{\pm 0.76}$ | $\mathbf{95.06}_{\pm 0.29}$ | $93.45_{\pm 0.92}$ | $72.68_{\pm 12.83}$ | $41.33_{\pm 6.72}$ |
| 20 | $\mathbf{48.79}_{\pm 0.27}$ | $47.96_{\pm 1.84}$ | $48.65_{\pm 0.91}$ | $47.71_{\pm 0.91}$ | $\mathbf{92.40}_{\pm 0.38}$ | $84.28_{\pm 1.35}$ | $63.84_{\pm 3.45}$ | $20.80_{\pm 0.99}$ |

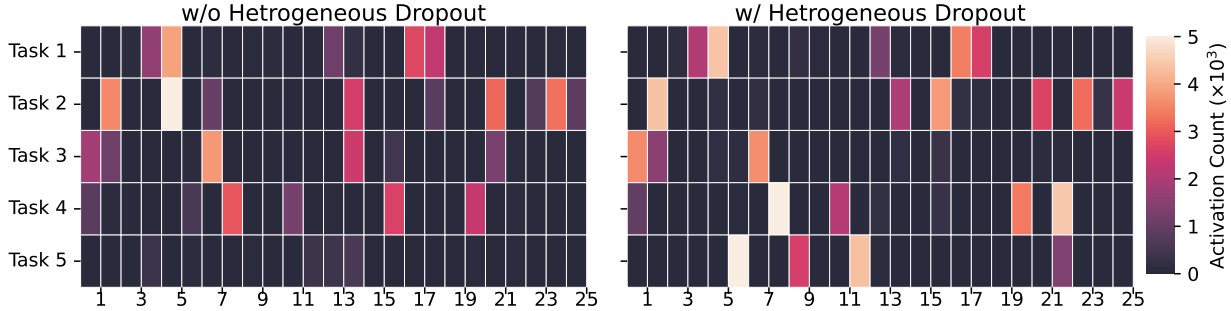

Figure 2: Total activation counts for the test set of each task (y-axis) for a random set of 25 units in the second hidden layer of the model. Heterogeneous dropout reduces the overlap in activations and facilitates the formation of task-specific subnetworks.

## 4.2 Sparse Activations Facilitate the Formation of Subnetworks

Neocortical circuits are characterized by high levels of sparsity in neural activations (Barth & Poulet, 2012; Graham & Field, 2006). There is further evidence suggesting that neuronal coding of natural sensory stimuli should be sparse (Barth & Poulet, 2012; Tolhurst et al., 2009). This is in stark contrast to the dense and highly entangled connectivity in standard ANNs. Particularly for CL, sparsity provides several advantages: sparse non-overlapping representations can reduce interference between tasks (Abbasi et al., 2022; Iyer et al., 2022; Aljundi et al., 2019), can lead to the natural emergence of task-specific modules (Hadsell et al., 2020).

We study the effect of different levels of activation sparsity by varying the ratio of active neurons in k-winner-take-all (k-WTA) activations (Ahmad & Scheinkman, 2019). Each hidden layer of our model has a constant sparsity in its connections (randomly 50% of weights are set to 0 at initialization) and propagates only the top-k activations (in Figure 1, k-WTA layer). Table 2 shows that sparsity plays a critical role in enabling CL in DNNs. Sparsity in activations effectively reduces interference by reducing the overlap in representations. Interestingly, the stark difference in the effect of different levels of sparse activations on Rot-MNIST and Perm-MNIST highlights the importance of considering task similarity in the design of CL methods. As the tasks in Perm-MNIST are independent of each other, having fewer active neurons (5%) enables the network to learn non-overlapping representations for each task, while the high task similarity in Rot-MNIST can benefit from overlapping representations, which allows for the reusability of features across the tasks. The number of tasks the agent has to learn also has an effect on the optimal sparsity level. In Appendix, we show that having different levels of sparsity in different layers can further improve performance. As the earlier layers learn general features, having a higher ratio of active neurons can enable higher reusability and forward transfer. For the later layers, a smaller ratio of active neurons can reduce interference between task-specific features.

## 4.3 Heterogeneous Dropout for Non-overlapping Activations and Subnetworks

Information in the brain is encoded by the strong activation of a relatively small set of neurons that form a sparse coding. A different subset of neurons is utilized to represent different types of stimuli Graham

Table 3: Effect of Heterogeneous dropout with increasing $\rho$ values on different datasets with a varying number of tasks.

| Dataset | # Tasks | w/o Dropout | Dropout parameter ($\rho$) | | | | |
|---|---|---|---|---|---|---|---|
| | | | 0.1 | 0.3 | 0.5 | 0.7 | 1.0 |
| Rot-MNIST | 5 | $92.28_{\pm 0.20}$ | $91.79_{\pm 0.53}$ | $92.53_{\pm 0.11}$ | $92.74_{\pm 0.38}$ | $93.19_{\pm 0.32}$ | $\mathbf{93.42_{\pm 0.25}}$ |
| | 10 | $70.78_{\pm 0.23}$ | $71.53_{\pm 1.07}$ | $72.38_{\pm 1.44}$ | $73.63_{\pm 1.00}$ | $74.20_{\pm 0.78}$ | $\mathbf{75.50_{\pm 0.74}}$ |
| | 20 | $48.79_{\pm 0.27}$ | $48.57_{\pm 0.90}$ | $48.91_{\pm 0.65}$ | $49.84_{\pm 0.59}$ | $51.03_{\pm 0.31}$ | $\mathbf{51.11_{\pm 0.76}}$ |
| Perm-MNIST | 5 | $95.77_{\pm 0.33}$ | $95.70_{\pm 0.29}$ | $95.97_{\pm 0.44}$ | $96.40_{\pm 0.28}$ | $\mathbf{96.58_{\pm 0.17}}$ | $96.48_{\pm 0.26}$ |
| | 10 | $95.06_{\pm 0.29}$ | $95.23_{\pm 0.04}$ | $95.65_{\pm 0.20}$ | $95.54_{\pm 0.26}$ | $95.74_{\pm 0.22}$ | $\mathbf{95.94_{\pm 0.24}}$ |
| | 20 | $92.40_{\pm 0.38}$ | $92.83_{\pm 0.42}$ | $\mathbf{93.20_{\pm 0.32}}$ | $92.82_{\pm 0.06}$ | $93.09_{\pm 0.47}$ | $91.77_{\pm 0.30}$ |

& Field (2006). Furthermore, there is evidence of non-overlapping representations in the brain. To mimic this, we employ Heterogeneous dropout (Abbasi et al., 2022) which in conjunction with context-dependent processing of information, can effectively reduce the overlap of representations, leading to less interference between tasks and, thereby, less forgetting. During training, we track the frequency of activations for each neuron in a layer for a given task, and in the subsequent tasks, the probability of a neuron being dropped is inversely proportional to its activation counts. This encourages the model to learn the new task using neurons that have been less active for previous tasks. Figure 1 shows that neurons that have been more active (darker shade) are more likely to be dropped before k-WTA is applied. Specifically, let $[a_t^l]_j$ denote the activation counter of the neuron $j$ in the layer $l$ after learning $t$ tasks. For the learning task $t + 1$, the probability that this neuron is retained is given by:

$$[p_{t+1}^l]_j = exp(\frac{-[a_t^l]_j}{\max_j [a_t^l]_j}\rho) \tag{5}$$

where $\rho$ controls the strength of enforcement of non-overlapping representations, with larger values leading to less overlap. This provides us with an efficient mechanism for controlling the degree of overlap between the representations of different tasks and, hence, the degree of forward transfer and interference based on the task similarities.

Table 3 shows that employing Heterogeneous dropout can further improve the performance of the model. We also analyze the effect of the $\rho$ parameter on the activation counts and the overlap in the representations. Figure 2 shows that Heterogeneous dropout can facilitate the formation of task-specific subnetworks and Figure 3 shows the symmetric KL-divergence between the distribution of activation counts on the test set of Task 1 and Task 2 on the model trained with different $\rho$ values on Perm-MNIST with two tasks. As we increase the $\rho$ parameter, the activations in each layer become increasingly dissimilar. Heterogeneous dropout provides a simple mechanism for balancing the reusability and interference of features depending on the similarity of tasks.

## 4.4 Layerwise Heterogeneous Dropout and Task Similarity

For an effective CL agent, it is important to maintain a balance between forward transfer and interference across tasks. As the earlier layers learn general features, a higher portion of the features can be reused to learn the new task, which can facilitate forward transfer, whereas the later layers learn more task-specific features, which can cause interference. Heterogeneous dropout provides us with an efficient mechanism for controlling the degree of overlap between the activations, and hence the features of each layer. Here, we investigate whether having different levels of sparsity (controlled with the $\rho$ parameter) in different layers can further improve performance. As the earlier layers learn general features, having higher overlap (smaller $\rho$) between the set of active neurons can enable higher reusability and forward transfer. For the later layers, a lesser overlap between the activations (higher $\rho$) can reduce interference between task-specific features.

To study the effect of Heterogeneous dropout in relation to task similarity, we vary the incremental rotation, $\theta_{inc}$, in each subsequent task for Rot-MNIST setting with 5 tasks. The rotation of task $\tau$ is given by $(\tau -$

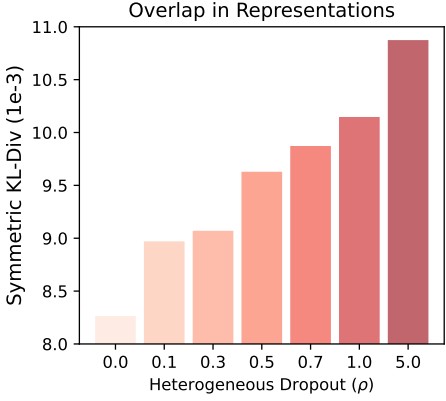
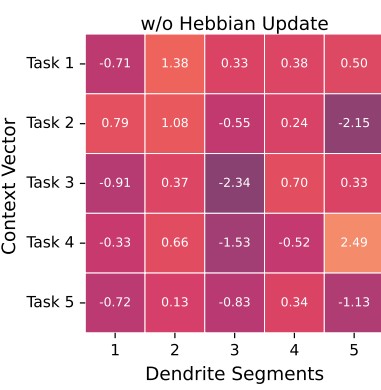
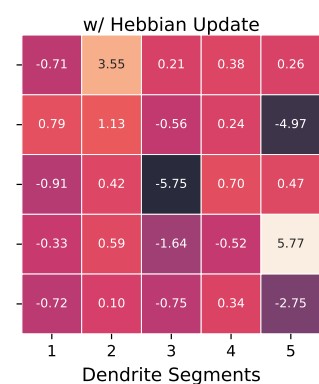

Figure 3: Effect of dropout $\rho$ on the overlap between the distributions of layer two activation counts for each task in Perm-MNIST with 2 tasks. Higher $\rho$ reduces the overlap.

Figure 4: Dendritic segment activations of a unit in layer 1 for the context vectors of each task for a model trained on Perm-MNIST with 5 tasks. Hebbian update strengthens the association between the context and the dendritic segments, increasing the magnitude of the modulating signal.

Table 4: Effect of layer-wise dropout $\rho$ on Rot-MNIST with 5 tasks with varying degrees of incremental rotation ($\theta_{inc}$) in each subsequent task. Row 0 shows ($\rho^{l_1}$, $\rho^{l_2}$) the $\rho$ values for the first and second hidden layers, respectively.

| $\rho^{l_1}$, $\rho^{l_2}$ | Task Similarity ($\theta_{inc}$) | | | | | |
|---|---|---|---|---|---|---|
| | 2 | 4 | 8 | 16 | 24 | 32 |
| 0.1, 0.1 | $97.60_{\pm 0.12}$ | $96.74_{\pm 0.16}$ | $91.79_{\pm 0.53}$ | $74.99_{\pm 1.16}$ | $63.33_{\pm 1.15}$ | $57.39_{\pm 2.36}$ |
| 0.1, 0.5 | $97.77_{\pm 0.08}$ | $97.02_{\pm 0.11}$ | $92.39_{\pm 0.39}$ | $75.56_{\pm 1.46}$ | $64.18_{\pm 1.79}$ | $57.05_{\pm 2.13}$ |
| 0.5, 0.5 | $97.88_{\pm 0.12}$ | $97.22_{\pm 0.11}$ | $92.74_{\pm 0.38}$ | $76.73_{\pm 1.01}$ | $64.18_{\pm 1.42}$ | $58.35_{\pm 0.73}$ |
| 0.5, 1.0 | $97.88_{\pm 0.04}$ | $\mathbf{97.25}_{\pm 0.11}$ | $92.87_{\pm 0.39}$ | $76.87_{\pm 0.39}$ | $64.84_{\pm 0.65}$ | $59.40_{\pm 2.31}$ |
| 1.0, 1.0 | $\mathbf{97.89}_{\pm 0.09}$ | $97.19_{\pm 0.09}$ | $93.42_{\pm 0.25}$ | $77.48_{\pm 0.94}$ | $66.33_{\pm 1.62}$ | $61.35_{\pm 1.90}$ |
| 1.0, 2.0 | $97.68_{\pm 0.09}$ | $97.00_{\pm 0.23}$ | $93.46_{\pm 0.78}$ | $79.07_{\pm 0.67}$ | $68.20_{\pm 2.34}$ | $63.08_{\pm 0.86}$ |
| 2.0, 2.0 | $97.42_{\pm 0.17}$ | $97.00_{\pm 0.11}$ | $\mathbf{93.53}_{\pm 0.53}$ | $80.03_{\pm 0.62}$ | $69.99_{\pm 1.97}$ | $65.74_{\pm 1.21}$ |
| 2.0, 5.0 | $97.39_{\pm 0.03}$ | $96.54_{\pm 0.15}$ | $92.95_{\pm 0.01}$ | $\mathbf{80.55}_{\pm 0.89}$ | $\mathbf{73.74}_{\pm 0.21}$ | $69.46_{\pm 2.66}$ |
| 5.0, 5.0 | $96.86_{\pm 0.11}$ | $96.12_{\pm 0.08}$ | $92.33_{\pm 0.18}$ | $79.53_{\pm 0.42}$ | $72.47_{\pm 1.23}$ | $\mathbf{70.77}_{\pm 2.11}$ |
| w/o Dropout | $97.72_{\pm 0.29}$ | $96.93_{\pm 0.40}$ | $92.31_{\pm 0.56}$ | $75.67_{\pm 1.40}$ | $63.68_{\pm 1.36}$ | $56.49_{\pm 2.86}$ |

1)$\theta_{inc}$. Table 4 shows the performance of the model for different layerwise $\rho$ values. Generally, heterogeneous dropout consistently improves the performance of the model, especially when the task similarity is low. For $\theta_{inc} = 32$, it provides $\sim 25\%$ improvement. As task similarity decreases ($\theta_{inc}$ increases), higher values of $\rho$ are more effective. Furthermore, we see that having different $\rho$ values for each layer can provide additional gains in performance.

## 4.5 Hebbian Learning Strengthens Context Gating

For a biologically plausible ANN, it is important to incorporate not only the design elements of biological neurons but also the learning mechanisms it employs. Lifetime plasticity in the brain generally follows the Hebbian principle: a neuron that consistently contributes to the firing of another neuron will build a stronger connection to that neuron (Hebb, 2005).

Therefore, we follow the approach in Flesch et al. (2023) to complement error-based learning with the Hebbian update to strengthen the connections between contextual information and dendritic segments (Figure 1(b)).

Each supervised parameter update with backpropagation is followed by a Hebbian update step on the dendritic segments to strengthen the connections between the context input and the corresponding dendritic segment, which is activated. To constrain the parameters, we use Oja's rule, which adds weight decay to Hebbian learning (Oja, 1982),

$$u_\kappa \leftarrow u_\kappa + \eta_h d(c - d u_\kappa) \tag{6}$$

where $\eta_h$ is the learning rate, $\kappa$ is the index of the winning dendrite with weight $u_\kappa$ and the modulating signal $d = u_\kappa^T c$ for the context signal $c$. Figure 4 shows that the Hebbian update step increases the magnitude of the modulating signal from the dendrites on the feedforward activity, which can further strengthen context-dependent gating and facilitate the formation of task-specific subnetworks. Table 1 shows that this can consequently lead to improvement in results. Though the gains are not considerable in these settings, Table 5 shows the gains with Hebbian Learning are more pronounced in challenging CL settings with higher task and dataset complexity.

## 4.6 Synaptic Consolidation Further Mitigates Forgetting

In addition to their integrative properties, dendrites also play a key role in retaining information and providing protection against erasure (Cichon & Gan, 2015; Yang et al., 2009). New spines that are formed on different sets of dendritic branches in response to learning different tasks are protected from being eliminated through mediation of synaptic plasticity and structural changes that persist when learning a new task (Yang et al., 2009).

We employ synaptic consolidation by incorporating *Synaptic Intelligence* (Zenke et al., 2017) (details in Appendix) which maintains an importance estimate of each synapse in an online manner during training and subsequently reduces the plasticity of synapses which are considered important for learned tasks. In particular, we adjust the importance estimate to account for the disparity in the degree to which updates to different parameters affect the output of the layer due to the inhibitory interneuron architecture in the DANN layers (Cornford et al., 2020). The importance estimates of the excitatory connections to the inhibitory units and the intra-layer inhibitory connections are upscaled to further penalize changes to these weights. Table 1 shows that employing Synaptic Intelligence (+SC) in this manner further mitigates forgetting. Particularly for Rot-MNIST with 20 tasks, it provides a considerable performance improvement.

## 4.7 Experience Replay is Essential for Enabling CL in Challenging Scenarios

Replay of past neural activation patterns in the brain is considered to play a critical role in memory formation, consolidation, and retrieval (Walker & Stickgold, 2004; McClelland et al., 1995). The replay mechanism in the hippocampus (Kumaran et al., 2016) has inspired a series of rehearsal-based approaches (Li & Hoiem, 2017; Chaudhry et al., 2019; Lopez-Paz & Ranzato, 2017; Arani et al., 2022) that have been proven to be effective in challenging continual learning scenarios (Farquhar & Gal, 2018; Hadsell et al., 2020). Therefore, to replay samples from previous tasks, we utilize a small episodic memory buffer that is maintained through *Reservoir sampling* (Vitter, 1985). It attempts to approximately match the distribution of the incoming stream by assigning equal probabilities to each new sample to be represented in the buffer. During training, samples from the current task, $(x_b, y_b) \sim \mathcal{D}_\tau$, are interleaved with memory buffer samples, $(x_m, y_m) \sim \mathcal{M}$ to approximate the joint distribution of tasks seen so far. Furthermore, to mimic the replay of the activation patterns that accompanied the learning event in the brain, we also save the output logits, $z_m$, across the training trajectory and enforce consistency loss when replaying samples from episodic memory. Concretely, the loss is given by:

$$\mathcal{L} = \mathcal{L}_{cls}(f(x_b; \theta), y_b) + \alpha \mathcal{L}_{cls}(f(x_m; \theta), y_m) + \beta(f(x_m; \theta) - z_m)^2 \tag{7}$$

where $f(.; \theta)$ is the model parameterized by $\theta$, $\mathcal{L}_{cls}$ is the standard cross-entropy loss, and $\alpha$ and $\beta$ controls the strength of interleaved training and the consistency constraint, respectively.

Table 1 shows that experience replay (+ER) complements context-dependent information processing and enables the model to learn the joint distribution well in varying challenging settings. In particular, the failure of the model to avoid forgetting in the Class-IL setting (Seq-MNIST) without experience replay

suggests that context-dependent processing of information alone does not suffice, and experience replay might be essential. Adding consistency regularization (+CR) further improves performance as the model receives additional relational information about the structural similarity of classes, which facilitates the consolidation of information.

Table 5: Effect of each component of the biologically plausible framework on different Domain-IL and Class-IL settings. For all experiments, we use a memory budget of 500 samples. We provide the average task performance and 1 std of 5 runs.

| Method | Domain-IL | Class-IL | | |
|---|---|---|---|---|
| | Rot-FMNIST | Seq-FMNIST | Seq-MNIST | Seq-GCIFAR10 |
| Joint | $98.15_{\pm 0.09}$ | $94.33_{\pm 0.51}$ | $94.53_{\pm 0.53}$ | $36.10_{\pm 0.86}$ |
| SGD | $51.89_{\pm 0.27}$ | $19.83_{\pm 0.04}$ | $19.83_{\pm 0.04}$ | $14.86_{\pm 1.06}$ |
| ActiveDANN | $49.42_{\pm 0.83}$ | $21.46_{\pm 1.95}$ | $19.97_{\pm 0.29}$ | $16.12_{\pm 0.31}$ |
| ActiveDANN + ER | $80.99_{\pm 0.53}$ | $77.56_{\pm 0.27}$ | $86.88_{\pm 0.83}$ | $28.74_{\pm 0.42}$ |
| + Hebbian Update | $82.16_{\pm 0.26}$ | $78.02_{\pm 0.38}$ | $88.39_{\pm 0.78}$ | $30.12_{\pm 0.49}$ |
| + SC | $82.55_{\pm 0.37}$ | $78.05_{\pm 0.61}$ | $88.79_{\pm 0.49}$ | $30.45_{\pm 0.61}$ |
| + HD | $83.97_{\pm 0.46}$ | $78.74_{\pm 0.38}$ | $89.60_{\pm 0.73}$ | $30.29_{\pm 0.51}$ |
| Bio-ANN | $\mathbf{89.22}_{\pm 0.21}$ | $\mathbf{79.28}_{\pm 0.42}$ | $\mathbf{89.90}_{\pm 0.24}$ | $\mathbf{30.95}_{\pm 0.17}$ |

## 4.8 Combining the individual components

Having shown the individual effect of each of the brain-inspired components in the biologically plausible framework, here we look at their combined effect. The resultant model is referred to as Bio-ANN. Table 1 shows that the different components complement each other and consistently improve the performance of the model. Our empirical results suggest that employing multiple complementary components and learning mechanisms, similar to the brain, may be an effective approach to enable continual learning in ANNs.

## 4.9 Additional Results on Challenging CL settings

To further evaluate the versatility of the biological components on more challenging settings, we conducted experiments on Fashion-MNIST and grayscale version of CIFAR10. We considered both the Class-IL and Domain-IL settings. Seq-FMNIST, Seq-MNIST, and Seq-GCIFAR10 divide the classification into 5 tasks with 2 classes each, while Rot-FMNIST involves 20 tasks with each task involving classifying the 10 classes in each task with the samples rotated by increments of 8 degrees.

For brevity, we refer to Active Dendrites + Dale's principle as ActiveDANN. To show the effect of different components better (ActiveDANN without ER fails in the class-IL setting), we consider ActiveDann + ER as the baseline upon which we add the other components. Empirical results in Table 5 show that the findings on MNIST settings also translate to more challenging datasets and each component leads to performance improvement. In particular, we observe that for more complex datasets, Hebbian learning provides a significant performance improvement. The preliminary results suggest that the effect of biological mechanisms and architecture might be more pronounced on more complex datasets and CL settings.

## 5 Discussion

Continual learning is a hallmark of intelligence, and the human brain constitutes the most efficient learning agent capable of CL. Therefore, incorporating the different components and mechanisms employed in the brain and studying their interactions can provide valuable insights for the design of ANNs suitable for CL. While there are several studies that are inspired by the brain, they focus primarily on one aspect. Since the brain employs all these different components in tandem, it stands to reason that their interactions, or complementary nature, are what enables effective learning instead of one component alone. Furthermore, the underlying framework within which these components are employed and the learning mechanisms might

also be critical. The effort to close the gap between current AI and human intelligence could benefit from our enhanced understanding of the brain and incorporating similar mechanisms in ANNs. This is the fundamental question we aimed to study and bring to the attention of the research community at large.

We conducted a study on the effect of different brain-inspired mechanisms under a biologically plausible framework in the CL setting. The underlying model incorporates several key components of the design principles and learning mechanisms in the brain: each layer constitutes separate populations of exclusively excitatory and inhibitory units, which adheres to Dale's principle, and the excitatory pyramidal neurons are augmented with dendritic segments for context-dependent processing of information. We first showed that equipped with the integrative properties of dendrites, the feedforward network adhering to Dale's principle perform on par with standard ANNs, and provide considerable performance gains in cases where the the quality of context signal in high. Then we studied the individual role of different components. We showed that controlling the sparsity in activations using k-WTA activations and Heterogeneous dropout mechanism that encourages the model to use a different set of neurons for each task is an effective approach for maintaining a balance between reusability of features and interference, which is critical for enabling CL. We further showed that complementing error-based learning with the "fire together, wire together" learning paradigm can further strengthen the association between the context signal and dendritic segments that process them and facilitate context-dependent gating. To further mitigate forgetting, we incorporated synaptic consolidation in conjunction with experience replay and showed their effectiveness in challenging CL settings. Finally, the combined effect of these components suggests that similar to the brain, employing multiple complementary mechanisms in a biologically plausible architecture is an effective approach to enable CL in ANN. It also provides a framework for further study of the role of inhibition in mitigating catastrophic forgetting.

However, there are several limitations and potential avenues for future research. In particular, as dendritic segments provide an effective mechanism for studying the effect of encoding different information in the context signal, they provide an interesting research avenue as to what information is useful for the sequential learning of tasks and the effect of different context signals. Neuroscience studies suggest that multiple brain regions are involved in processing a stimulus and, while there is evidence that active dendritic segments receive contextual information that is different from the input received by the proximal segments, it is unclear what information is encoded in the contextual information and how it is extracted. Here, we used the context signal as in (Iyer et al., 2022), which aims to encode the identity of the task by taking the average input image of all the samples in the task. Although this approach empirically works well in the Perm-MNIST setting, it is important to consider its utility and limitations under different CL settings. Given the specific design of Perm-MNIST, binary-centered digits, and the independent nature of the permutations in each task, the average input image can provide a good approximation of the applied permutation, and hence efficiently encode the task identity. However, this is not straightforward for Rot-MNIST where the task similarities are higher and even more challenging for natural images, where averaging the input image does not provide a meaningful signal. More importantly, it does not seem biologically plausible to encode task information alone as the context signal and ignore the similarity of classes occurring in different tasks. For instance, it seems more reasonable to process slight rotations of the same digits similarly (as in Rot-MNIST) rather than processing them through different subnetworks. This argument is supported by the performance degradation on the Rotated-MNIST setting with active dendrites over standard ANNs. Ideally, we would like the context signal for different rotations of a digit to be highly similar. It is, however, quite challenging to design context signals that can capture a wide range of complexities in the sequential learning of tasks. Furthermore, instead of hand engineering the context signal to bias learning towards certain types of tasks, an effective approach for learning the context signal in an end-to-end training framework is an interesting direction for future search. Another limitation of our framework would be the use of backpropagation which is widely criticized for being biologically implausible as it requires symmetric weight matrices in the feedforward and feedback pathways Xiao et al. and requires each neuron to have access to and contribute towards optimizing a single global objective function. The synapses in the brain, on the other hand, are unidirectional, and feedforward and feedback connections are physically distinct. The brain is also believed to perform localized learning. An interesting avenue for future research could be to extend our framework with more biologically plausible learning rules and study how the learning rule affects the interactions between the biologically plausible

mechanisms. Future studies can also study the role of more plausible organization of neurons instead of fully connected, for instance, incorporation of population coding.

In general, our study presents a compelling case for incorporating the design principles and learning machinery of the brain into ANNs and provides credence to the argument that distilling the details of the learning machinery of the brain can bring us closer to human intelligence (Hassabis et al., 2017; Hayes et al., 2021). Furthermore, deep learning is increasingly being used in neuroscience research to model and analyze brain data (Richards et al., 2019). The utility of the model for such research depends on two critical aspects: the performance of the model and how close the architecture is to the brain (Cornford et al., 2020; Schrimpf et al., 2020). The biologically plausible framework in our study incorporates several design components and learning mechanisms of the brain and performs well in a (continual learning) task that is closer to human learning. Therefore, we believe that this work may also be useful for the neuroscience community in evaluating and guiding computational neuroscience. Studying the properties of ANNs with higher similarity to the brain may provide insight into the mechanisms of brain functions. We believe that the fields of artificial intelligence and neuroscience are intricately intertwined, and progress in one can drive the other as well.

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

# A    Appendix

## A.1    Related Work - Biological Inspired AI

The human brain has long been a source of inspiration for ANNs design (Hassabis et al., 2017; Kudithipudi et al., 2022). However, we have failed to take full advantage of our enhanced understanding of the brain, and there are fundamental differences between the design principles and learning mechanisms employed in the brain and ANNs. These differences may account for the huge gap in performance and behavior.

From an architectural design perspective, standard ANNs are predominantly based on the point neuron model (Lapique, 1907) which is an outdated and oversimplified model of biological computations that lacks sophisticated and context-dependent processing in the brain. Furthermore, neurons in standard ANNs lack adherence to Dale's principle (Strata et al., 1999) to which most neurons in the brain adhere. Unlike the brain where presynaptic neurons have an exclusively excitatory or inhibitory impact on their postsynaptic partners, neurons in standard ANNs contain both positive and negative output weights, and signs can change while learning. These constitute two of the major fundamental differences in the underlying design principle of ANNs and the brain.

Two recent studies attempt to address this gap. Cornford et al. (2020) incorporated Dale's principle into ANNs (DANNs) in a more biologically plausible manner and show that with certain initialization and regularization considerations, DANNs can perform comparable to standard ANNs in object recognition tasks, which earlier attempts failed to do so. Our study extends DANNs to the more challenging CL setting and shows that accounting for the discrepancy in the effect of weight changes in excitatory and inhibitory neurons can further reduce forgetting in CL. Iyer et al. (2022) propose an alternative to the point neuron model and provide an algorithmic abstraction of pyramidal neurons in the neocortex. Each neuron is augmented with dendritic segments which receive an additional context signal and the output of the dendrite segment modulates the activity of the neuron, allowing context-dependent processing of information. Our study builds upon their work and provides a biologically plausible architecture characterized by both adherence to Dale's principle and the context-dependent processing of pyramidal neurons. This provides us with a framework to study the role of brain-inspired mechanisms and allows us to study the role of inhibition in the challenging continual learning setting, which is closer to human learning.

From a learning perspective, several approaches have been inspired by the brain, particularly for CL (Kudithipudi et al., 2022). The replay mechanism in the hippocampus (Kumaran et al., 2016) has inspired a series of rehearsal-based approaches (Hayes et al., 2021; Hadsell et al., 2020) which have proven to be effective in challenging continual learning scenarios (Farquhar & Gal, 2018). Another popular approach for continual learning, regularization-based approaches Zenke et al. (2017); Kirkpatrick et al. (2017), has been inspired by neurobiological models that suggest that CL in the neocortex is based on a task-specific synaptic consolidation process that involves rendering a proportion of synapses less plastic and, therefore, stable over long timescales (Benna & Fusi, 2016; Yang et al., 2009). While both approaches are inspired by the brain, researchers have mostly discounted the fact that the brain employs both of them in conjunction to consolidate information rather than in isolation. Therefore, the research in both of these methods has been orthogonal. Furthermore, they have been applied on top of standard ANNs which are not representative of the complexities of the neuron in the brain. Our study employs replay and synaptic consolidation together in a more biologically plausible architecture and shows that they complement each other to improve performance.

Furthermore, our framework employs several techniques to mimic the characteristics of activations in the brain. As pyramidal neurons in the neocortex have highly sparse connectivity to each other (Hunter et al., 2021; Holmgren et al., 2003) and only a small percentage ($<2\%$) of neurons are active for given stimuli neurons (Barth & Poulet, 2012), we apply k-winner-take-all (k-WTA) activations (Ahmad & Scheinkman, 2019) to mimic activation sparsity. Several studies have shown the benefits of sparsity in CL (Abbasi et al., 2022; Mallya et al., 2018; Aljundi et al., 2019), they do not consider that the brain not only utilizes sparsity, it does so in an efficient manner to encode information. Information in the brain is encoded by the strong activation of a relatively small set of neurons, forming sparse coding. A different subset of neurons is utilized to represent different types of stimuli (Foldiak, 2003) and semantically similar stimuli activate a similar set of

neurons. Heterogeneous dropout (Abbasi et al., 2022) coupled with k-WTA activations aims to mimic these characteristics by encouraging the new task to utilize a new set of neurons for learning. Finally, we argue that it is important not only to incorporate the design elements of biological neurons but also the learning mechanisms that they employ. Lifetime plasticity in the brain generally follows the Hebbian principle (Hebb, 2005). Therefore, we follow the approach in Flesch et al. (2023) to complement error-based learning with Hebbian update to strengthen the connections between contextual information and dendritic segments and show that it strengthens context gating.

Our study provides a biologically plausible framework with the underlying architecture with the context-dependent processing of information and adherence to Dale's principle. Additionally, it employs the learning mechanisms (experience replay, synaptic consolidation, and Hebbian update) and characteristics (sparse non-overlapping activations and task-specific subnetworks) of the brain. To the best of our knowledge, we are the first to provide a comprehensive study of the integration of different brain-inspired mechanisms in a biologically plausible architecture in a CL setting.

## A.2 Training Complexity

Here we discuss the training complexity of the different components in our framework. While Dale Principle adds additional learning parameters, subtractive inhibition allows for efficient implementation as the effective weight can be represented as $W^l = W^l_{ee} - W^l_{ei} W^l_{ie}$. Also as the number of inhibitory neurons is substantially lower than excitatory neurons, the additional parameters are low. The computational cost associated with Active Dendrites is twofold: the cost of creating the context signal for the task and the subsequent computations for the modulating signal in dendritic segments. Both of these depend significantly on the specific approach for evaluating the context signal and the dimensions of the context vector. The implemented approach takes an average input image of all the training samples for a task and therefore requires an additional pass through the training samples and the dimension of the context vector is equal to the dimensions of the input image. More efficient approaches for creating context vectors would considerably reduce the computational cost associated with active dendrites. Activation sparsity, on the hand, substantially reduces the required computations for both the forward and backward pass (e.g. only 5% of the units are active in the forward pass, and therefore only these units will be trained) especially if compiler and hardware support sparse computations. Heterogeneous Dropout does not add a significant computation head as it only involves tracking the activation counts of the model and applying a masking operation during training. Synaptic Intelligence maintains an online estimate of parameter importance without requiring an additional forward pass. It does however keep a checkpoint of the model's previous weights for penalizing changes in weights.

We would like to emphasize that the brain employs a set of complex mechanisms in an efficient manner. As we bring more attention to employing similar mechanisms in ANNs and their complementary interactions, research in this promising direction would inevitably lead to more efficient and effective learning.

## A.3 Biological Components on Standard ANN

To further evaluate the applicability of the biological mechanisms in our framework and the role of underlying biologically plausible architecture, we conduct an ablation study on top of standard ANN. Table 6 shows the effect of adding activation sparsity using k-WTA to standard ANNs upon which then heterogeneous dropout (HD), synaptic consolidation (SC), and Experience Replay (ER) are added before finally combining all components (Standard-BioANN). Results show that activation sparsity considerably improves the performance on Perm-MNIST. However, it reduces the performance on Rot-MNIST which can be attributed to the low degree of active neurons or the lack of forward transfer as the task similarity is high. Adding the biological components provide consistent performance improvements similar to the original settings. The results are lower than ActiveDann experiments which provide further credence to the arguments that biological mechanisms on top of a biologically plausible architecture can be more effective.

Table 6: Effect of each component of the biologically plausible framework on top of standard ANN on different datasets with varying number of tasks. We provide the average task performance and 1 std of five runs.

| Method | Rot-MNIST | | | Perm-MNIST | | | Seq-MNIST |
|---|---|---|---|---|---|---|---|
| | 5 | 10 | 20 | 5 | 10 | 20 | |
| Joint | $98.27_{\pm0.07}$ | $98.25_{\pm0.06}$ | $98.17_{\pm0.11}$ | $97.64_{\pm0.19}$ | $97.61_{\pm0.15}$ | $97.59_{\pm0.08}$ | $94.53_{\pm0.50}$ |
| Standard ANN | $93.79_{\pm0.32}$ | $74.13_{\pm0.36}$ | $51.89_{\pm0.36}$ | $77.96_{\pm8.34}$ | $76.42_{\pm2.54}$ | $65.18_{\pm2.12}$ | $19.83_{\pm0.04}$ |
| + Activation Sparsity | $92.36_{\pm0.30}$ | $70.61_{\pm0.51}$ | $48.04_{\pm0.32}$ | $93.14_{\pm0.32}$ | $87.01_{\pm0.80}$ | $72.22_{\pm0.92}$ | $19.85_{\pm0.02}$ |
| + HD | $92.72_{\pm0.36}$ | $72.97_{\pm0.55}$ | $49.54_{\pm0.69}$ | $93.80_{\pm0.31}$ | $89.07_{\pm0.82}$ | $74.38_{\pm0.59}$ | $24.28_{\pm8.28}$ |
| + SC | $93.77_{\pm0.28}$ | $74.68_{\pm0.48}$ | $51.45_{\pm1.01}$ | $96.07_{\pm0.11}$ | $95.30_{\pm0.19}$ | $92.86_{\pm0.43}$ | $20.19_{\pm0.25}$ |
| + ER | $94.81_{\pm0.21}$ | $87.84_{\pm0.37}$ | $76.09_{\pm0.51}$ | $95.52_{\pm0.08}$ | $93.11_{\pm0.17}$ | $86.34_{\pm0.27}$ | $81.44_{\pm0.72}$ |
| Standard-Bio-ANN | $\mathbf{96.66}_{\pm0.09}$ | $\mathbf{93.32}_{\pm0.28}$ | $\mathbf{88.57}_{\pm0.50}$ | $\mathbf{96.72}_{\pm0.04}$ | $\mathbf{96.27}_{\pm0.05}$ | $\mathbf{95.35}_{\pm0.10}$ | $\mathbf{89.67}_{\pm0.46}$ |

Table 7: Forward Transfer with different dropout strength ($\rho$) on Rot-MNIST with 5 tasks with varying degrees of incremental rotation ($\theta_{inc}$) in each subsequent task. We report the average and standard deviation of 5 runs.

| $\rho^{l_1}, \rho^{l_2}$ | Task Similarity ($\theta_{inc}$) | | | | | |
|---|---|---|---|---|---|---|
| | 2 | 4 | 8 | 16 | 24 | 32 |
| w/o Dropout | $88.78_{\pm0.90}$ | $88.10_{\pm0.98}$ | $86.52_{\pm0.47}$ | $82.35_{\pm0.75}$ | $73.94_{\pm0.82}$ | $60.37_{\pm0.87}$ |
| 0.1, 0.1 | $88.85_{\pm0.75}$ | $88.05_{\pm0.81}$ | $86.99_{\pm0.78}$ | $82.45_{\pm0.88}$ | $73.30_{\pm0.34}$ | $60.03_{\pm1.35}$ |
| 0.5, 0.5 | $88.96_{\pm0.76}$ | $88.31_{\pm0.91}$ | $87.11_{\pm0.80}$ | $82.33_{\pm0.82}$ | $73.66_{\pm0.54}$ | $59.94_{\pm1.64}$ |
| 1.0, 1.0 | $88.92_{\pm0.83}$ | $88.10_{\pm0.86}$ | $86.89_{\pm0.75}$ | $81.97_{\pm0.89}$ | $72.80_{\pm0.52}$ | $59.54_{\pm0.96}$ |
| 5.0, 5.0 | $88.38_{\pm0.83}$ | $87.62_{\pm0.92}$ | $85.73_{\pm0.96}$ | $78.76_{\pm0.93}$ | $67.75_{\pm1.16}$ | $53.65_{\pm1.07}$ |

## A.4   Forward and Backward Transfer

To further understand the effect of modularity on the stability and plasticity of the model, we evaluate the degree of forward transfer and forgetting (Chaudhry et al., 2018) in the model depending on the task similarity and the degree of modularity enforced through different strengths of heterogeneous dropout. Forward transfer assesses whether the model is capable of improving on unseen tasks w.r.t. random guessing, whereas forgetting measures the performance degradation in subsequent tasks. We evaluate forgetting as the average difference between the maximum performance of the model on a task (when it is learned) and the final accuracy at the end of training on all tasks. We vary the incremental rotation, $\theta_{inc}$, in each subsequent task for Rot-MNIST setting with 5 tasks to simulate different degrees of task similarity with higher $\theta_{inc}$ corresponding to more dissimilar tasks. The degree of modularity is controlled with different strengths of heterogeneous dropout ($\rho$ parameter) with higher $\rho$ values corresponding to lower overlap in representations and hence higher modularity.

Table 7 provides the forward transfer while Table 8 provides the backward transfer. We observe that generally as the tasks become more dissimilar, forward transfer reduces and there is more forgetting. Additionally, increasing the modularity (higher $\rho$ values) generally leads to less forward transfer while reducing forgetting. Heterogeneous dropout provides us with an efficient mechanism to control the plasticity and stability of the model and an optimal balance between the stability and plasticity can provide the best performance. Interestingly, we observe that in certain higher similarity cases, moderate level of modularity sometimes increases the forward transfer.

## A.5   Experimental Setup

To study the role of the different components inspired by the brain in a biologically plausible NN for CL and to gauge their roles in the performance and characteristics of the model, we conduct all our experiments

Table 8: Degree of Forgetting with different dropout strength ($\rho$) on Rot-MNIST with 5 tasks with varying degrees of incremental rotation ($\theta_{inc}$) in each subsequent task. We report the average and standard deviation of 5 runs.

| $\rho^{l_1}$, $\rho^{l_2}$ | Task Similarity ($\theta_{inc}$) | | | | | |
|---|---|---|---|---|---|---|
| | 2 | 4 | 8 | 16 | 24 | 32 |
| w/o Dropout | $0.47_{\pm0.16}$ | $1.30_{\pm0.25}$ | $7.10_{\pm0.60}$ | $29.96_{\pm3.04}$ | $43.06_{\pm2.61}$ | $50.96_{\pm1.96}$ |
| 0.1, 0.1 | $0.38_{\pm0.14}$ | $1.19_{\pm0.29}$ | $6.50_{\pm1.55}$ | $26.87_{\pm1.58}$ | $42.03_{\pm0.46}$ | $50.45_{\pm1.07}$ |
| 0.5, 0.5 | $0.13_{\pm0.14}$ | $0.81_{\pm0.31}$ | $6.12_{\pm1.28}$ | $25.93_{\pm2.00}$ | $41.63_{\pm1.71}$ | $46.78_{\pm2.08}$ |
| 1.0, 1.0 | $0.12_{\pm0.10}$ | $0.61_{\pm0.19}$ | $5.52_{\pm1.04}$ | $23.77_{\pm0.71}$ | $38.29_{\pm1.39}$ | $44.36_{\pm2.43}$ |
| 5.0, 5.0 | $0.61_{\pm0.25}$ | $1.46_{\pm0.22}$ | $5.81_{\pm0.06}$ | $21.82_{\pm1.37}$ | $27.39_{\pm1.11}$ | $30.95_{\pm1.83}$ |

Table 9: The selected hyperparamneters for the experiments showing the individual effect of each component (Table 1). The base learning rate for all the experiments is 0.3 and the individual components use the same learning rates for $W_{ie}$ and $W_{ei}$ as (+ Dale's principle). For + SC experiments, we use $\lambda_{W_{ie}}$=10 and $\lambda_{W_{ei}}$=10. For ER experiments, we use a memory budget of 500 samples.

| Dataset | #Tasks | + Dale's Principle | + Hebbian Update | + SC | + ER | + ER + CR |
|---|---|---|---|---|---|---|
| Rot-MNIST | 5 | $\eta_{W_{ie}}$=3e-2, $\eta_{W_{ei}}$=3e-3 | $n_h$=3e-10 | $\lambda$=0.25 | $\alpha$=1, $\beta$=0 | $\alpha$=1, $\beta$=0.50 |
| | 10 | $\eta_{W_{ie}}$=3e-2, $\eta_{W_{ei}}$=3e-3 | $n_h$=3e-08 | $\lambda$=0.25 | $\alpha$=1, $\beta$=0 | $\alpha$=1, $\beta$=0.50 |
| | 20 | $\eta_{W_{ie}}$=3e-3, $\eta_{W_{ei}}$=3e-4 | $n_h$=3e-10 | $\lambda$=1.00 | $\alpha$=1, $\beta$=0 | $\alpha$=1, $\beta$=0.50 |
| Perm-MNIST | 5 | $\eta_{W_{ie}}$=3e-2, $\eta_{W_{ei}}$=3e-2 | $n_h$=3e-09 | $\lambda$=0.10 | $\alpha$=1, $\beta$=0 | $\alpha$=1, $\beta$=0.50 |
| | 10 | $\eta_{W_{ie}}$=3e-2, $\eta_{W_{ei}}$=3e-2 | $n_h$=3e-06 | $\lambda$=0.25 | $\alpha$=1, $\beta$=0 | $\alpha$=1, $\beta$=0.50 |
| | 20 | $\eta_{W_{ie}}$=3e-2, $\eta_{W_{ei}}$=3e-3 | $n_h$=3e-09 | $\lambda$=0.10 | $\alpha$=1, $\beta$=0 | $\alpha$=1, $\beta$=0.50 |
| Seq-MNIST | 5 | $\eta_{W_{ie}}$=3e-2, $\eta_{W_{ei}}$=3e-3 | $n_h$=3e-07 | $\lambda$=0.25 | $\alpha$=1, $\beta$=0 | $\alpha$=1, $\beta$=0.25 |

under uniform settings. Unless otherwise stated, we use a multi-layer perception (MLP) with two hidden layers with 2048 units and k-WTA activations. Each neuron is augmented with $N$ dendritic segments where $N$ corresponds to the number of tasks and the dimensions correspond to the dimensions of the context vector that correspond to the input image size (784 for all MNIST-based settings). The model is trained using an SGD optimizer with 0.3 learning rate and a batch size of 128 for 3 epochs on each task. We set the weight sparsity to 0 and set the percentage of active neurons to 5%. For our experiments involving Dale's principle, we maintain the same number of total units in each layer divided into 1844 excitatory and 204 inhibitory units. Only the excitatory units are augmented with dendritic segments. Importantly, we use the initialization strategy and corrections for the SGD update as posited in Cornford et al. (2020) to account for the disparities in the degree to which updates to different parameters affect the layer output distribution. Updates to inhibitory unit parameters are scaled down relative to update of excitatory parameters. Concretely, the gradient updates to $W_{ie}$ were scaled by $\sqrt{n_e}^{-1}$ and $W_{ei}$ by $d^{-1}$, where $n_e$ are the number of excitatory neurons in the layer and d is the input dimension of the layer. Furthermore, to select the hyperparameters for different settings, we use a small validation set. Note that, as the goal was not to achieve the best possible accuracy, but rather to show the effect of each component, we did not conduct an extensive hyperparameter search. Table 9 provides the selected hyperparameters for the effect of individual component experiments in Table 1 and Table 10 provides the selected hyperparameter for Bio-ANN experiments. We report the mean accuracy over all tasks and 1 std over three different random seeds.

## A.6 Effect of adjusting for the inhibitory weights

The inhibitory interneuron architecture of DANN layers introduces disparities in the degree to which updates to different parameters affect the layer's output distribution, e.g. if a single element of $W_{ie}$ is updated, this

Table 10: The selected hyperparamneters for Bio-ANN experiments in Table 1. We use the same learning rate for each setting as + Dale's principle (Table 9).

| Dataset | #Tasks | $n_h$ | $\lambda$ | $\rho$ | $\alpha$ | $\beta$ |
|---------|--------|-------|-----------|--------|----------|---------|
| Rot-MNIST | 5 | 3e-8 | 0.25 | 0.1 | 1 | 0.5 |
| | 10 | 3e-8 | 0.1 | 0.3 | 1 | 0.5 |
| | 20 | 3e-8 | 0.1 | 0.3 | 1 | 0.5 |
| Perm-MNIST | 5 | 3e-6 | 0.1 | 0.1 | 1 | 0.5 |
| | 10 | 3e-8 | 0.1 | 0.3 | 1 | 0.5 |
| | 20 | 3e-8 | 0.1 | 0.3 | 1 | 0.5 |
| Seq-MNIST | 5 | 3e-6 | 0.1 | 0.1 | 1 | 0.25 |

Table 11: Effect of adjusting the learning rates of $W_{ie}$ and $W_{ei}$ at the end of the first task on different datasets with a varying number of tasks.

| $\eta_{Wie}$ | $\eta_{Wei}$ | Rot-MNIST | | | Perm-MNIST | | |
|--------------|--------------|-----------|-----------|-----------|------------|-----------|-----------|
| | | 5 | 10 | 20 | 5 | 10 | 20 |
| 3e-1 | 3e-1 | $92.12_{\pm 0.34}$ | $70.86_{\pm 0.44}$ | $46.30_{\pm 1.03}$ | $95.78_{\pm 0.19}$ | $94.73_{\pm 0.36}$ | $\mathbf{92.67}_{\pm 0.61}$ |
| 3e-2 | 3e-2 | $92.23_{\pm 0.53}$ | $70.23_{\pm 1.12}$ | $47.53_{\pm 1.79}$ | $95.77_{\pm 0.33}$ | $\mathbf{95.06}_{\pm 0.29}$ | $91.63_{\pm 0.39}$ |
| 3e-2 | 3e-3 | $92.28_{\pm 0.27}$ | $70.78_{\pm 0.23}$ | $47.32_{\pm 1.43}$ | $95.68_{\pm 0.14}$ | $94.96_{\pm 0.49}$ | $92.40_{\pm 0.38}$ |
| 3e-3 | 3e-3 | $\mathbf{92.34}_{\pm 0.51}$ | $\mathbf{71.27}_{\pm 1.69}$ | $47.81_{\pm 1.10}$ | $95.70_{\pm 0.29}$ | $94.44_{\pm 0.70}$ | $92.02_{\pm 0.19}$ |
| 3e-3 | 3e-4 | $92.03_{\pm 0.09}$ | $70.79_{\pm 1.75}$ | $\mathbf{48.79}_{\pm 0.27}$ | $\mathbf{95.90}_{\pm 0.13}$ | $94.19_{\pm 0.47}$ | $91.04_{\pm 0.34}$ |

has an effect on each element of the layer's output. An inhibitory weight update of $\delta$ to $W_{ie}$ changes the model distribution approximately $n_e$ times more than an excitatory weight update of $\delta$ to $W_{ee}$ (Cornford et al., 2020). The effect of these disparities would be even more pronounced in a CL setting as large changes to the output distribution when learning a new task can cause more forgetting of previous tasks. To account for these, we further reduce the learning rate of $W_ie$ and $W_ei$ after learning the first task. Table 11 shows that accounting for the higher effect of inhibitory neurons can further improve the performance of the model in majority of the settings. It would be interesting to explore better approaches to account for the aforementioned disparities, which are tailored for CL and consider the effect on forgetting.

### A.7 Effect of adjusting for scaling the SI importance estimate for inhibitory weights

Similar to adjusting the learning rate of inhibitory weights, we check whether scaling up the importance estimate of inhibitory neurons can further improve the effectiveness of synaptic consolidation in reducing forgetting. Table 12 shows that scaling the importance estimate in accordance with the degree to which inhibitory weights affect the output distribution, and hence forgetting, further improves the performance in majority of cases, especially for a lower number of tasks. This suggests that regularization methods designed specifically for networks with inhibitory neurons is a promising research direction.

### A.8 Effect of Sparsity

To further study the effect of different levels of sparsity in activations and connections, we vary the number of weights randomly set to zero at initialization ($S_W \in \{0, 0.25, 0.50\}$) and the ratio of active neurons ($k^{l1}, k^{l2}$) in each hidden layer. Table 13 shows that sparsity in activation plays a critical role in enabling CL in ANNs. Interestingly, sparsity in connections play a considerable role in Perm-MNIST with higher levels of active neurons ($\geq 0.2$). Furthermore, exploring finer differences in activation sparsity of different layers may further improve the performance. Similar to heterogeneous dropout, we show the effect of activation sparsity in relation to task similarity in Table 14. Similar tasks (lower $\theta_{inc}$) benefit from a higher number of active

Table 12: Effect of scaling the importance estimate for $W_{ie}$ and $W_{ei}$ to reduce the parameter shift in the inhibitory weights on Rot-MNIST and Perm-MNIST datasets with varying number of tasks.

| $\lambda$ | $\lambda_{W_{ie}}$ | $\lambda_{W_{ei}}$ | Rot-MNIST | | | Perm-MNIST | | |
|---|---|---|---|---|---|---|---|---|
| | | | 5 | 10 | 20 | 5 | 10 | 20 |
| | 1 | 1 | $92.92_{\pm0.22}$ | $75.53_{\pm1.46}$ | $52.16_{\pm1.17}$ | $96.59_{\pm0.17}$ | $96.20_{\pm0.13}$ | $\mathbf{95.69}_{\pm0.10}$ |
| 0.1 | 10 | 10 | $92.77_{\pm0.27}$ | $74.70_{\pm1.05}$ | $51.94_{\pm1.05}$ | $96.57_{\pm0.34}$ | $96.18_{\pm0.18}$ | $95.64_{\pm0.15}$ |
| | 10 | 100 | $92.50_{\pm0.65}$ | $74.67_{\pm1.27}$ | $52.55_{\pm1.12}$ | $\mathbf{96.67}_{\pm0.23}$ | $96.26_{\pm0.26}$ | $95.61_{\pm0.10}$ |
| | 1 | 1 | $92.77_{\pm0.69}$ | $\mathbf{76.30}_{\pm0.77}$ | $55.05_{\pm2.47}$ | $96.62_{\pm0.28}$ | $96.07_{\pm0.02}$ | $95.36_{\pm0.24}$ |
| 0.25 | 10 | 10 | $\mathbf{93.54}_{\pm0.79}$ | $75.44_{\pm0.81}$ | $54.93_{\pm2.35}$ | $96.54_{\pm0.22}$ | $96.23_{\pm0.16}$ | $95.03_{\pm0.13}$ |
| | 10 | 100 | $93.40_{\pm0.86}$ | $75.87_{\pm1.35}$ | $55.06_{\pm1.77}$ | $96.65_{\pm0.25}$ | $\mathbf{96.36}_{\pm0.10}$ | $95.18_{\pm0.29}$ |
| | 1 | 1 | $93.23_{\pm0.71}$ | $75.24_{\pm0.62}$ | $\mathbf{60.24}_{\pm2.02}$ | $96.22_{\pm0.37}$ | $95.97_{\pm0.06}$ | $92.94_{\pm0.85}$ |
| 0.5 | 10 | 10 | $93.13_{\pm0.24}$ | $75.95_{\pm0.38}$ | $59.35_{\pm1.53}$ | $96.24_{\pm0.51}$ | $95.81_{\pm0.10}$ | $92.86_{\pm0.87}$ |
| | 10 | 100 | $92.85_{\pm0.42}$ | $75.85_{\pm1.18}$ | $58.94_{\pm2.15}$ | $96.36_{\pm0.42}$ | $96.02_{\pm0.27}$ | $93.48_{\pm0.57}$ |
| w/o SC | | | $92.28_{\pm0.27}$ | $70.78_{\pm0.23}$ | $48.79_{\pm0.27}$ | $95.77_{\pm0.33}$ | $95.06_{\pm0.29}$ | $92.40_{\pm0.38}$ |

Table 13: Effect of different levels of sparsity in activations (ratio of active neurons, $(k^{l1}, k^{l2})$ in $1^{st}$ and $2^{nd}$ hidden layer respectively) and connections (ratio of zero weights, $S_W$) on Rot-MNIST and Perm-MNNIST with increasing number of tasks. The best performance across the different sparsity levels for each task is in bold.

| | #Tasks | $S_W$ | Activation Sparsity $(k^{l1}, k^{l2})$ | | | | | | |
|---|---|---|---|---|---|---|---|---|---|
| | | | (0.05, 0.05) | (0.1, 0.05) | (0.1, 0.1) | (0.2, 0.1) | (0.2, 0.2) | (0.5, 0.2) | (0.5, 0.5) |
| Rot-MNIST | 5 | 0.00 | $92.28_{\pm0.27}$ | $92.63_{\pm0.46}$ | $92.26_{\pm0.31}$ | $92.25_{\pm0.71}$ | $\mathbf{92.79}_{\pm0.44}$ | $92.51_{\pm0.50}$ | $92.26_{\pm0.65}$ |
| | | 0.25 | $91.22_{\pm0.58}$ | $91.50_{\pm0.36}$ | $92.33_{\pm0.26}$ | $91.75_{\pm0.30}$ | $92.60_{\pm0.19}$ | $91.23_{\pm1.63}$ | $91.66_{\pm0.46}$ |
| | | 0.50 | $90.78_{\pm0.11}$ | $91.15_{\pm0.34}$ | $91.25_{\pm0.14}$ | $91.25_{\pm0.92}$ | $90.67_{\pm0.73}$ | $90.75_{\pm0.86}$ | $90.41_{\pm1.48}$ |
| | 10 | 0.00 | $70.78_{\pm0.23}$ | $71.16_{\pm0.51}$ | $71.95_{\pm1.54}$ | $72.22_{\pm0.63}$ | $\mathbf{73.32}_{\pm0.69}$ | $71.89_{\pm0.62}$ | $71.61_{\pm0.76}$ |
| | | 0.25 | $70.23_{\pm0.76}$ | $71.42_{\pm0.98}$ | $71.84_{\pm0.72}$ | $73.22_{\pm1.34}$ | $72.58_{\pm0.64}$ | $72.23_{\pm0.71}$ | $71.23_{\pm0.84}$ |
| | | 0.50 | $69.61_{\pm0.50}$ | $70.59_{\pm0.62}$ | $70.94_{\pm0.71}$ | $72.05_{\pm0.34}$ | $72.25_{\pm1.40}$ | $71.37_{\pm0.65}$ | $70.85_{\pm0.67}$ |
| | 20 | 0.00 | $\mathbf{48.79}_{\pm0.27}$ | $48.01_{\pm0.58}$ | $47.96_{\pm1.84}$ | $48.33_{\pm1.23}$ | $48.65_{\pm0.91}$ | $48.19_{\pm0.14}$ | $47.71_{\pm0.91}$ |
| | | 0.25 | $47.72_{\pm0.83}$ | $48.61_{\pm0.30}$ | $48.41_{\pm0.84}$ | $48.53_{\pm1.77}$ | $48.30_{\pm0.87}$ | $48.29_{\pm1.59}$ | $47.11_{\pm0.44}$ |
| | | 0.50 | $46.20_{\pm0.26}$ | $47.15_{\pm1.37}$ | $48.02_{\pm1.10}$ | $48.17_{\pm1.42}$ | $48.30_{\pm1.41}$ | $47.66_{\pm1.53}$ | $47.73_{\pm0.51}$ |
| Perm-MNIST | 5 | 0.00 | $95.77_{\pm0.33}$ | $95.55_{\pm0.27}$ | $\mathbf{96.43}_{\pm0.10}$ | $95.85_{\pm0.29}$ | $90.29_{\pm6.07}$ | $88.18_{\pm8.86}$ | $74.51_{\pm13.55}$ |
| | | 0.25 | $95.45_{\pm0.25}$ | $95.14_{\pm0.27}$ | $95.65_{\pm0.22}$ | $95.75_{\pm0.32}$ | $93.73_{\pm1.29}$ | $87.49_{\pm2.33}$ | $75.97_{\pm5.61}$ |
| | | 0.50 | $93.95_{\pm0.65}$ | $94.19_{\pm0.41}$ | $94.90_{\pm0.22}$ | $94.22_{\pm1.19}$ | $94.05_{\pm0.81}$ | $91.02_{\pm3.71}$ | $83.94_{\pm9.58}$ |
| | 10 | 0.00 | $\mathbf{95.06}_{\pm0.29}$ | $94.08_{\pm0.95}$ | $94.38_{\pm0.73}$ | $89.54_{\pm2.27}$ | $78.91_{\pm5.26}$ | $76.44_{\pm7.93}$ | $35.86_{\pm2.04}$ |
| | | 0.25 | $94.51_{\pm0.12}$ | $93.52_{\pm0.01}$ | $93.62_{\pm0.64}$ | $88.58_{\pm2.17}$ | $84.94_{\pm3.58}$ | $69.32_{\pm6.24}$ | $59.53_{\pm9.57}$ |
| | | 0.50 | $92.12_{\pm0.62}$ | $90.31_{\pm2.01}$ | $92.29_{\pm0.73}$ | $88.48_{\pm0.22}$ | $82.57_{\pm2.40}$ | $74.42_{\pm2.84}$ | $69.31_{\pm5.66}$ |
| | 20 | 0.00 | $\mathbf{92.40}_{\pm0.38}$ | $89.41_{\pm1.73}$ | $84.28_{\pm1.35}$ | $73.29_{\pm2.60}$ | $63.84_{\pm3.45}$ | $58.85_{\pm5.00}$ | $20.80_{\pm0.99}$ |
| | | 0.25 | $90.75_{\pm0.63}$ | $89.22_{\pm0.94}$ | $84.29_{\pm2.17}$ | $75.82_{\pm6.68}$ | $67.26_{\pm1.22}$ | $63.96_{\pm1.19}$ | $41.60_{\pm2.64}$ |
| | | 0.50 | $87.31_{\pm0.82}$ | $84.90_{\pm1.99}$ | $83.72_{\pm0.63}$ | $69.62_{\pm5.98}$ | $66.27_{\pm2.74}$ | $64.42_{\pm2.33}$ | $51.92_{\pm3.79}$ |

neurons that can increase forward transfer, while dissimilar tasks (higher $\theta_{inc}$) perform better with higher activation sparsity that can reduce the overlap in representations.

Table 14: Effect of different levels of activation sparsity on Rot-MNIST with 5 tasks with varying degrees of incremental rotation ($\theta_{inc}$) in each subsequent task. Row 0 shows ($k^{l_1}$, $k^{l_2}$) the ratio of active neurons in the $1^{st}$ and $2^{nd}$ hidden layers, respectively.

| ($k^{l_1}$, $k^{l_2}$) | Task Similarity ($\theta_{inc}$) | | | | | |
|---|---|---|---|---|---|---|
| | 2 | 4 | 8 | 16 | 24 | 32 |
| 0.05, 0.05 | $97.54_{\pm 0.06}$ | $96.56_{\pm 0.14}$ | $91.95_{\pm 0.54}$ | $75.13_{\pm 0.83}$ | $63.14_{\pm 0.52}$ | $57.01_{\pm 0.89}$ |
| 0.1, 0.05 | $97.57_{\pm 0.32}$ | $96.84_{\pm 0.18}$ | $92.49_{\pm 0.59}$ | $76.15_{\pm 1.28}$ | $64.03_{\pm 1.12}$ | $58.56_{\pm 1.17}$ |
| 0.1, 0.1 | $97.81_{\pm 0.08}$ | $96.84_{\pm 0.30}$ | $92.28_{\pm 0.31}$ | $\mathbf{76.80}_{\pm 1.20}$ | $\mathbf{64.91}_{\pm 1.17}$ | $\mathbf{58.62}_{\pm 1.70}$ |
| 0.2, 0.1 | $97.44_{\pm 0.74}$ | $96.88_{\pm 0.45}$ | $92.47_{\pm 0.69}$ | $75.79_{\pm 1.26}$ | $64.38_{\pm 1.58}$ | $58.34_{\pm 1.54}$ |
| 0.2, 0.2 | $\mathbf{97.88}_{\pm 0.11}$ | $\mathbf{97.27}_{\pm 0.15}$ | $\mathbf{92.79}_{\pm 0.50}$ | $76.22_{\pm 1.46}$ | $64.30_{\pm 1.38}$ | $57.22_{\pm 1.06}$ |
| 0.5, 0.2 | $97.67_{\pm 0.64}$ | $96.92_{\pm 0.78}$ | $92.61_{\pm 0.65}$ | $75.66_{\pm 0.95}$ | $63.86_{\pm 0.61}$ | $56.11_{\pm 1.20}$ |
| 0.5, 0.5 | $97.67_{\pm 0.55}$ | $97.03_{\pm 0.53}$ | $92.29_{\pm 0.58}$ | $74.55_{\pm 1.01}$ | $62.37_{\pm 0.62}$ | $53.13_{\pm 3.62}$ |

---

**Algorithm 1** Bio-ANN: A biologically plausible framework for CL.

---

**Input:** Data stream $\mathcal{D}$; Learning rates $\eta$, $\eta_{W_{ie}}$, $\eta_{W_{ei}}$; Hebbian learning rate $\eta_h$; Heterogeneous dropout $\rho$; Synaptic consolidation weights $\lambda$, $\lambda_{W_{ie}}$, $\lambda_{W_{ei}}$, $\gamma$; Experience replay weights $\alpha$, $\beta$

**Initialize:**

Model weights $\theta$, Reference weights $\theta_c = \{\}$, Task prototypes $C_\tau = \{\}$

Heterogeneous dropout: Overall activation counts $A_\tau = 0$, Keep probabilities $P_\tau = 1$

Memory buffer $\mathcal{M} \leftarrow \{\}$

Synaptic Intelligence: $\omega = 0$, $\Omega = 0$

$\triangleright$ Sample task from data stream

1: **for** $\mathcal{D}_\tau \in \{\mathcal{D}_1, \mathcal{D}_2, .., \mathcal{D}_T\}$ **do**

$\triangleright$ Task context

2:     Evaluate context vector (Eq. 2):

$c_\tau = \frac{1}{|\mathcal{D}_\tau|} \sum_{x \in \mathcal{D}_\tau} x$

3:     Update the set of prototypes:

$C_\tau \leftarrow \{C_\tau, c_\tau\}$

$\triangleright$ Train on task $\tau$

4:     **while** Training **do**

5:        Sample data: $(x_b, y_b) \sim \mathcal{D}_\tau$ and $(x_m, y_m, z_m) \sim \mathcal{M}$

$\triangleright$ Task specific loss

6:        Get the model output and activation counts on the current task batch:

$z_b, a_b = F(x_b, c_\tau; \theta, P_\tau)$     # Apply Heterogeneous dropout

7:        Calculate task loss:

$\mathcal{L}_\tau = \mathcal{L}_{cls}(z_b, y_b)$

8:        Update overall activation counts:

$A_\tau \leftarrow \text{UpdateActivationCounts}(a_t)$

$\triangleright$ Experience replay

9:        Infer context for buffer samples (Eq. 3):

$c_m = \arg\min_{c_\tau} \|\boldsymbol{x'} - \boldsymbol{C_\tau}\|_2$

10:       Get model output on buffer samples:

$z = F(x_m, c_m; \theta)$        # Disable Heterogeneous dropout

11:       Calculate replay loss:

$\mathcal{L}_{er} = \alpha \mathcal{L}_{cls}(z, y_m) + \beta(z - z_m)^2$

$\triangleright$ Synaptic regularization

12:       Calculate SI loss:

$\mathcal{L}_{sc} = \Omega_{adj}(\theta - \theta_c)^2$

13:       Calculate overall loss and clip the gradient between 0 and 1:

$\mathcal{L} = \mathcal{L}_\tau + \mathcal{L}_{er} + \mathcal{L}_{sc}$

$\nabla_\theta \mathcal{L} = \text{Clip}(\nabla_\theta \mathcal{L}, 0, 1)$

$\triangleright$ Update Models

14:       SGD update: $\theta = \text{UpdateModel}(\nabla_\theta \mathcal{L}, \eta, \eta_{W_{ie}}, \eta_{W_{ei}})$

15:       Hebbian update on dendritic segments:    $U = \text{HebbianStep}(\{c_\tau, c_m\}, U)$

16:

17:       Update small omega:    $\omega = \omega + \eta \nabla_\theta^2 \mathcal{L})$          $\triangleright$ Update SI parameter

18:       $\mathcal{M} \leftarrow Reservoir(\mathcal{M}, (x_b, y_b, z_b))$       $\triangleright$ Update memory buffer (Algorithm 2)

19:                                              $\triangleright$ Task Boundary

20:     Update keep Probabilities (Eq 5):    $P_\tau = exp(\frac{-A_\tau}{\max A_\tau} \rho)$

21:     Update SI Omega and reference weights and reset small omega:

$\Omega = \Omega + \frac{\omega}{(\theta - \theta_c)^2 + \gamma}$

$\omega = 0$

$\theta_c = \theta$

22:     Scale up importance for inhibitory weights

$\Omega_{adj} = \text{ScaleUpInhib}(\Omega, \lambda_{W_{ie}}, \lambda_{W_{ei}})$

**return** $\theta$

---

**Algorithm 2** Reservoir Sampling

**Input:** Memory Buffer $\mathcal{M}$, Buffer Size $\mathcal{B}$, Number of examples seen so far $N$, data points $(x, y, z)$

1: **if** $\mathcal{B} > N$ **then**                                                     $\triangleright$ Check if memory is full

2:    $\mathcal{M}[N] \leftarrow (x, y, z)$

3: **else**                                                         $\triangleright$ Select a sample to replace

4:    $n = \text{SampleRandomInteger}(\text{min}=0, \text{max}=N)$

5:    **if** $n < \mathcal{B}$ **then**

6:       $\mathcal{M}[n] \leftarrow (x, y, z)$

   **return** $\mathcal{M}$

