# OpenReview forum: "A Study of Biologically Plausible Neural Network: The Role and Interactions of Brain-Inspired Mechanisms in Continual Learning"
_TMLR — Accepted by TMLR_

### Review · Reviewer_kt2N · 2023-02-13

**Summary Of Contributions:**

The paper investigates how combinations of biologically plausible methods perform in continual learning. The authors use approaches proposed in previous works such as architectures based on Dale's principle, sparsity-enforcing activation functions (k-WTA), heterogeneous dropout, synaptic intelligence and experience replay. They test the listed methods on continual versions of the MNIST benchmark, focusing in particular on their combined performance. The paper also includes various ablation studies (e.g. impact of the task similarity on the correct choice of dropout temperature).

**Audience:**

Yes

**Broader Impact Concerns:**

I don't think the paper needs a broader impact statement.

**Claims And Evidence:**

Yes

**Requested Changes:**

I would like to ask the authors to address the weaknesses listed in the section above. However, as I mentioned, I think the paper is already useful as it is, so I'd be in favor of accepting the paper if the authors explicitly listed the issues above as limitations without introducing the changes.

Suggested improvements (corresponding to the list of weaknesses)
- Report metrics such as forward transfer in the study. Additionally, include an additional experiment checking transfer depending on task similarity and degree of modularity in the model.
- Include more combinations of the tested methods in the empirical study.
- Include a standard ANN as a baseline. Ideally, show how it performs with some of the suggested improvements (k-WTA, experience replay, Synaptic Intelligence).
- Extend results on Fashion-MNIST. Ideally, run some experiments on datasets such as CIFAR-10, even with linear (rather than convolutional) layers.
- Include more results on Incremental Class tasks. Incremental Task would also be a nice addition, but it's usually regarded as easier than Incremental Domain so it's probably not necessary.

**Strengths And Weaknesses:**

In general, although I think the paper is certainly valuable for the community and the experiments seem mostly sound, there are several drawbacks and limitations that need addressing.


Strengths:
- I think the topic of this study is interesting and useful for the community. As the authors note, many continual learning approaches are biologically inspired and the field has strong connections to neurobiology. However, there are few studies that thoroughly discuss these links and in particular the way different methods work when combined. As such, I think this paper should be of interest to TMLR's audience.
- Authors consider many different methods of various types in their study (e.g. regularization-based Synaptic Intelligence, sparsity-enforcing k-WTA and rehearsal-based experience replay). I appreciate that since many papers only focus on a single family of methods in their investigations.
- The paper includes many ablation studies showing how different values of hyperparameters impact the results, allowing the reader to build a better intuition of the problem and the performance of the tested approaches.
- The paper is well written and nicely discusses related work. I appreciate the detailed discussion and the extensive related work section in the appendix.

Weaknesses:
- Many of the methods tested in this paper enforce modularity in some ways (e.g. sparsity-enforcing k-WTA). This of course limits forgetting as it reduces interference, but at the same time it raises the question of whether the network is able to reuse the knowledge between these modules. As such, I don't think the issue of transfer between different tasks has been analyzed well enough in this paper. In my opinion, the question of "can we efficiently reuse prior knowledge" should be treated in continual learning with equal importance as the question "can we avoid forgetting?", and the paper mostly focuses on the second one.
- Investigating the performance of the combinations of methods is interesting, but at the same time, I'm not sure if the paper sufficiently covers the space of possible ways how to combine them. In particular, Table 1 only contains the base ActiveDANN model combined with additional methods. It would be interesting to see the synergies and incompatibilities of different approaches (e.g. what if we only use Synaptic Intelligence and Experience Replay). I raise this issue as checking combinations of different methods seems to be a major point of the paper.
- Although the study mostly focuses on biologically-plausible approaches, I think it should also consider the standard baselines used in continual learning such as a standard MLP trained with SGD. Such an approach is only tested in Table 2 in the Appendix and then it is only compared to biologically plausible approaches which include ER, which I think is an unfair comparison. Given these points, I don't think the following statement in Discussion is true: "We first showed that equipped with the integrative properties of dendrites, the feedforward network adhering to Dale’s principle not only performs as well as standard ANNs, but also provides gains."
- The empirical evaluation is mostly conducted using the MNIST benchmark. In continual learning literature, and in deep learning in general, it is no longer considered an interesting benchmark by itself -- many methods that perform well on MNIST do not scale well to tasks better representative of the real world. The authors do include some Fashion-MNIST results in the appendix, but by itself, I still think this is not enough. I do understand that the presented method might not work well with convolutional networks so it might be difficult to show interesting results on e.g. CIFAR-10. Still, given the experimental protocols in continual learning, I have to say that the datasets used here are too simplistic.
- Also, the authors mostly consider the Incremental Domain setting. There is a single experiment with the Incremental Class scenario in Table 1, but it seems that Experience Replay is the only approach that really works here. This is a limitation in the scope of the study.

Additional comment: The paper focuses on studying how to combine the strengths of previously proposed biologically-plausible continual learning methods, rather than proposing new solutions. This is not a weakness, especially given TMLR's guidelines, but it's worth noting.

---

> ### Author Response · Authors · 2023-03-02
> **Author's Response to Reviewer kt2N (1/2)**
>
> Thank you for your encouraging remarks, we are glad that you feel our study will be valuable for the community. We appreciate your valuable suggestions and feedback.
>
> > Report metrics such as forward transfer in the study. Additionally, include an additional experiment checking transfer depending on task similarity and degree of modularity in the model.
>
> Thank you for your insightful comment and for bringing this to our attention, forward transfer is indeed an important metric for a CL algorithm. We added a section on forward and backward transfer (Section A.4) where we study the effect of modularity on the stability and plasticity of the model. we evaluate the degree of forward transfer and backward transfer in the model depending on the task similarity and the degree of modularity enforced through different strengths of heterogeneous dropout. We vary the incremental rotation, in each subsequent task for Rot-MNIST setting with 5 tasks to simulate different degrees of task similarity with higher increment corresponding to more dissimilar tasks. The degree of modularity is controlled with different strengths of heterogeneous dropout with higher $\rho$ values corresponding to lower overlap in representations and hence higher modularity.
>
> Table 7 provides the forward transfer while Table 8 provides the backward transfer. We observe that generally as the tasks become more dissimilar, forward transfer reduces and there is more forgetting. Additionally, increasing the modularity generally leads to less forward transfer while reducing forgetting. Heterogeneous dropout provides us with an efficient mechanism to control the plasticity and stability of the model and an optimal balance between them can provide the best performance. Interestingly, we observe that in certain higher similarity cases, moderate level of modularity sometimes increases the forward transfer.
>
> > Include more combinations of the tested methods in the empirical study.
>
> > Extend results on Fashion-MNIST. Ideally, run some experiments on datasets such as CIFAR-10, even with linear (rather than convolutional) layers.
>
> > Include more results on Incremental Class tasks. Incremental Task would also be a nice addition, but it's usually regarded as easier than Incremental Domain so it's probably not necessary.
>
> Thank you for the valuable suggestions. We have added additional results on more challenging CL scenarios (Section 4.9) which aims to incorporate these suggestions.
>
> we conducted additional experiments on Fashion-MNIST, Seq-MNIST and the gray scale version of CIFAR10. Seq-FMNIST, Seq-MNIST, Seq-GCIFAR10 divides the classification into 5 tasks with 2 classes each. We use an additional hidden layer for Seq-GCIFAR10 experiments as the dataset is more complex.
> .
> For brevity, we refer to Active Dendrites + Dale’s principle as ActiveDANN. To show the effect of different components better (ActiveDANN without ER fails in the class-IL setting), we consider ActiveDann + ER as the baseline upon which we add the other components. This also adds more combinations of the biological components (ER + Hebb, ER + Synaptic Consolidation, ER + Heterogeneous Dropout).
>
> Empirical results in Table 4.7 show that the findings on MNIST settings also translate to more challenging datasets and each component leads to performance improvement. In particular, we observe that for more complex datasets, hebbian learning provides significant performance improvement. The preliminary results suggest that the effect of biological mechanisms and architecture might be more pronounced on more complex datasets and CL settings.
>
> > Include a standard ANN as a baseline. Ideally, show how it performs with some of the suggested improvements (k-WTA, experience replay, Synaptic Intelligence).
>
> We have also added lower and upper bounds for all the settings in Table 2 and ran the experiments for two additional seeds. We observe that for the Rotated-MNIST setting, active dendrites actually reduce the performance over SGD. This can be attributed to the quality of the context signal for Rotated-MNIST and the inability of an average image to provide useful information in this setting which we had highlighted in our discussion. This is in stark contrast with Permuted-MNIST where Active dendrites significantly improve the CL performance.
>
> Furthermore, we conducted ablation experiments on Standard-ANNs in Section A.3, incorporating activation sparsity, heterogeneous dropout, synaptic consolidation, and experience replay. We found that adding these biological components consistently improved performance, although the results were not as strong as those from ActiveDann experiments. This supports our argument that biological mechanisms, when combined with a biologically plausible architecture, can be highly effective.

---

> ### Author Response · Authors · 2023-03-02
> **Author's Response to Reviewer kt2N (2/2)**
>
> We kindly refer the reviewer to our general response for an overview of the changes made in the revised manuscript. We hope that our response adequately addresses your primary concerns. If you require additional information, please let us know. We would be happy to engage in further discussion.

---

### Review · Reviewer_PmJe · 2023-02-14

**Summary Of Contributions:**

The authors  consider a biologically plausible framework that constitutes separate populations of exclusively excitatory and inhibitory neurons that adhere to Dale’s principle, and the excitatory pyramidal neurons are augmented with dendritic-like structures for context-dependent processing of stimuli. They conduct a comprehensive study on the role and interactions of different mechanisms
inspired by the brain, including sparse non-overlapping representations, Hebbian learning, synaptic consolidation, and replay of past activations that accompanied the learning event. Their study suggests that the employing of multiple complementary mechanisms in a biologically plausible architecture, similar to the brain, may be effective in enabling continual learning in ANNs.

**Audience:**

Yes

**Claims And Evidence:**

Yes

**Requested Changes:**

Please provide a discussion on the weaknesses highlighted above.

**Strengths And Weaknesses:**

+ Very interesting framework and the authors have shown the effectiveness in different CL MNIST benchmarks
+ The bio-plausibility of the method points to some interesting learning behaviors that is not shown with previous CL methods.

Some of my comments on the work are:
- What is the complexity of training in the author's approach? A body of CL works today [1, 2] also try to look into how much memory compression and how much cost is being used to train the model in a CL manner. A discussion on that ll be helpful.
[1[ Venkatesha Y, Kim Y, Park H, Li Y, Panda P. Addressing Client Drift in Federated Continual Learning with Adaptive Optimization. Available at SSRN 4188586. 2022 Mar 24.
[2] Saha G, Garg I, Roy K. Gradient projection memory for continual learning. arXiv preprint arXiv:2103.09762. 2021 Mar 17.

- How does the dendritic connections addition/removal work in terms of training complexity? If your network is changing, then does that mean you are reinitializing the model, or starting from a previously stored point?

- Finally, I was wondering if the authors can comment on how will their approach scale with larger datasets or increasingly complex CL tasks.

---

> ### Author Response · Authors · 2023-03-02
> **Author's Response to Reviewer PmJe**
>
> Thank you for your positive feedback and insightful questions. We hope to address your concerns below
>
> > What is the complexity of training in the author's approach?
>
> The training complexity is indeed an important aspect of a CL algorithm and we have added a discussion on the training complexity of the model (Section A.2 in the Appendix.). We would like to emphasize that the brain employs a set of complex mechanisms in an efficient manner. As we bring more attention to employing similar mechanisms in ANNs and their complementary interactions, research in this promising direction would inevitably lead to more efficient and effective learning.
>
> > How does the dendritic connections addition/removal work in terms of training complexity? If your network is changing, then does that mean you are reinitializing the model, or starting from a previously stored point?
>
> The computational cost associated with Active Dendrites is twofold: the cost of creating the context signal for the task and the subsequent computations for the modulating signal in dendritic segments. Both of these depend significantly on the specific approach for evaluating the context signal and the dimensions of the context vector. The implemented approach takes an average input image of all the training samples for a task and therefore requires an additional pass through the training samples and the dimension of the context vector is equal to the dimensions of the input image. More efficient approaches for creating context vectors would considerably reduce the computational cost associated with active dendrites.
>
> The Dendrites segments provide us with a mechanism to enable the context-dependent processing of information. They are not comparable to adding additional learning parameters in the neural network or reinitializing the model as they perform a vastly different function, i.e. processing an additional context vector and modulating the activity of the feedforward neurons. The structure of the network does not change in the course of training for our experiment. Please let us know if we misunderstood your question, we would be happy to provide further details and clarifications.
>
>
>
> > Finally, I was wondering if the authors can comment on how will their approach scale with larger datasets or increasingly complex CL tasks.
>
> As our aim was to study the role of individual components and further our understanding of the complementary nature of the components working together, we wanted to keep the model simple and make minimal modifications to the original architectures used in DANN and Active Dendrites.
>
> We believe that both of these architectures can be extended to convolutional neural networks for more complex vision tasks. DANN discusses ideas on how their work can be extended to CNNs. For Active Dendrites, we can consider each convolutional kernel as a kernel and modulate the activations map based on the dendrite segments associated with each filter and the context signal. It would be interesting to see future work extend the framework to CNNs and more complex datasets. We believe that the advantages of context-dependent processing and inhibitory neurons would be enhanced for complex datasets.
>
> We conducted preliminary experiments on more challenging CL scenarios (Section 4.9). To further evaluate the versatility of the biological components on more challenging settings, we added more experiments on the challenging Class-IL and also considered the gray scale version of CIFAR10. Empirical results in Table 5 show that the findings on MNIST settings also translate to more challenging datasets and each component leads to significant performance improvement. In particular, we observe that for more complex datasets, hebbian learning provides significant performance improvement. The preliminary results suggest that the effect of biological mechanisms and architecture might be more pronounced on more complex datasets and CL settings.
>
> We would like to refer the reviewer to our general response for an overview of the changes in the revised manuscript. We hope that our response addresses your main concerns. Please let us know if we can provide further information. We would be happy in engage in a discussion.

---

### Review · Reviewer_baVv · 2023-02-15

**Summary Of Contributions:**

The authors explore the effect of combining various biologically plausible architectural features on the ability of the network to continually learn. Specifically, they combine active dendrites, Dale's principle, k-WTA activations, heterogeneous dropout (both layer-wise and otherwise), Hebbian learning, Synaptic consolidation (using synaptic intelligence), experience replay with consistency regularization. They find that adding some of these features can lead to improvements in the continual learning ability of the network.

**Audience:**

Yes

**Broader Impact Concerns:**

None.

**Claims And Evidence:**

No

**Requested Changes:**

I would request the following changes to make the manuscript stronger and more focused.
1. Run more than 3 runs for each feature.
2. Report baseline with ANN, add ablation studies of various features without active dendrites as well.
3. Either an explanation should be given for using features that don't contribute to CL performance or they should be
   discarded.
4. The use of backprop should be more prominently mentioned and discussed.

Minor:
- Table 1: Please add explanations for the various acronyms or descriptions of what each column (5, 10, 20) denotes.
- Fig. 1(a): arrows are missing on the green line, $W_{ie}$ and $W_{ei}$ seem to be referring to the same connection.
- In section 4.3, not clear what the Table 3 results contribute over those already in (Abbasi et al. 2022). Please
    explain or remove redundant results.

**Strengths And Weaknesses:**

### Strengths

The authors study the effect of combining various biologically plausible features, which to my knowledge hasn't been
done before. It is also useful to see that various different biologically plausible features don't interfere with each
other and prevent CL.

The authors also do extensive experiments, which provides a lot of information on which features are important,
although the analysis is not fully thorough -- see below.

The candid discussion of biological background on the use of contextual information is very insightful and informative.

### Weaknesses

Overall, the conclusions of and the outcomes from this work are not clear. The authors try out a number of features, but
   some of them don't clearly contribute to improved performance in the continual learning (CL) task and some of them do. Is the
   goal to add all known biological mechanisms and say that CL is still possible? Or is the goal to isolate those features
   that actually make a different in CL tasks? The focus of the paper requires sharpening.

1. Baseline performance on a ANN without any of the features is not compared (e.g. Table 1).
2. The study of all the features such as Hebbian Updates, synaptic consolidation etc. using a network with active
   dendrites as the basis. But it would have been useful to see the effect of these features on CL performance even
   without active dendrites to know if they absolutely require active dendrites.
3. There are some potentially biologically implausible mechanisms which are not discussed in sufficient detail -- the use of back-propagation for instance.
4. It's not clear if layer-wise heterogeneous dropout and Hebbian learning actually contribute to any improvement in CL
   performance overall.
5. For a study which focuses on studying the importance of various features/mechanisms for CL, reporting the task
   performance over only 3 runs makes it really hard to judge how robustly these mechanisms contribute to the CL
   performance.

Minor:
- Why are 90% inhibitory neurons used instead of the more widespread and presumably more biologically supported 80%?

---

> ### Author Response · Authors · 2023-03-02
> **Author's Response to Reviewer baVv (1/2)**
>
> Thank you for your valuable feedback and suggestions. We are glad that you found the study useful and the discussion insightful. We hope to have
>
> > the conclusions of and the outcomes from this work are not clear.
>
> We would like to restate that the goal of our study is to further our understanding of how the interactions between the different components and mechanisms employed in the brain and their complementary nature can provide valuable insights for the design of ANNs. Therefore the primary goal is to not only gauge the usefulness of a mechanism based on performance improvement in CL scenarios but also study the characteristics instilled in the model. Our key takeaway from the study is to show that biological components instill several desirable characteristics in the model which work in synergy with each other and can collectively lead to better performance. Another contribution of our work would be the biologically plausible framework that can be further studied and extended by future work.
>
> > The use of backprop should be more prominently mentioned and discussed.
>
> We would like to thank the reviewer for pointing out the biologically implausible mechanisms in the framework. We have extended the discussion section and highlighted the biological implausibility of backpropagation and the organization of neurons in the fully connected network. We also highlight some potential avenues that future work can contribute towards.
>
> > Run more than 3 runs for each feature.
>
> As per the reviewer's suggestions, we have run experiments for additional seeds and reported an average and std of 5 seeds in Table 1. The findings of the study still hold and we hope that these improve the statistical significance of our results.
>
> > Baseline performance on a ANN without any of the features is not compared.
>
> We have also added lower and upper bounds for all the settings. We observe that for the Rotated-MNIST setting, active dendrites actually reduce the performance over SGD. This can be attributed to the quality of the context signal for Rotated-MNIST and the inability of an average image to provide useful information in this setting which we had highlighted in our discussion. This is in stark contrast with Permuted-MNIST where Active dendrites significantly improve the CL performance.
>
> > Either an explanation should be given for using features that don't contribute to CL performance or they should be discarded.
>
> We believe that the biological component do contribute to the performance and instills desirable characteristics in the model. For instance, while the performance improvement with Hebbian learning might not be considerable, we see that it strengthens context gating which might be a critical element of the learning repository of the brain. In our additional experiments on more complex datasets (Section 4.9), we observe that the performance improvements are more pronounced. The preliminary results on more complex settings suggest that the effect of biological mechanisms and architecture might be more pronounced on more complex datasets and CL settings
>
> > Report baseline with ANN, add ablation studies of various features without active dendrites as well.
>
> We have further added ablation experiments on Standard-ANNs (Section A.3) with activation sparsity, heterogeneous dropout, Synaptic Consolidation and Experience Replay. Adding the biological components provide consistent performance improvements similar to the original settings. The results are lower than ActiveDann experiments which provide further credence to the arguments that biological mechanisms on top of a biologically plausible architecture can be more effective.
>
> > Why are 90% inhibitory neurons used instead of the more widespread and presumably more biologically supported 80%?
>
> We followed the original work which uses 10% of the neurons as inhibitory. It is generally believed that there are far more (~5-10 times) excitatory neurons in the cortical regions of the brain than inhibitory neurons [1]. Our framework is flexible and any ratio of excitatory to inhibitory neurons can be employed. Future work can further study the effect of the percentage of inhibitory neurons on the performance and characteristics of the model.
>
> > Table 1: Please add explanations for the various acronyms or descriptions of what each column (5, 10, 20) denotes
>
> We apologize for the confusion. We explicitly mentioned in Table 1 that these numbers correspond to the number of tasks.

---

> > ### Comment · Reviewer_baVv · 2023-03-27
> > **Updates make the paper stronger**
> >
> > Thank you for your response and updates. The empirical claims is much more solid now. One last comment:
> >
> > > Therefore the primary goal is to not only gauge the usefulness of a mechanism based on performance improvement in CL scenarios but also study the characteristics instilled in the model.
> >
> > I don't see an explicit study of any other characteristics instilled in the model other than the performance, although that would definitely have been quite interesting to see.

---

> > > ### Author Response · Authors · 2023-04-03
> > > **Author's Response**
> > >
> > > Thank you for your encouraging feedback. We are glad that you consider the empirical claims stronger.
> > > In addition to the performance, we aimed to study additional characteristics instilled in the model which include the formation of task-specific subnetworks (Figure 2), overlap in activations (Figure 3 Left), and the strengthening of context gating (Figure 3 Right). We further studied the effect of dropout strength for varying degrees of Task similarity (Table 4).  In the Appendix, we have also added a study on the forward and backward transfer of the model. We hope that these analyses provide additional insights into the model's behavior.
> > >
> > > We would appreciate suggestions on any further analysis we can perform to study the characteristics of the model more explicitly. Thank you for your valuable time and insights.

---

> ### Author Response · Authors · 2023-03-02
> **Author's Response to Reviewer baVv (2/2)**
>
> >In section 4.3, not clear what the Table 3 results contribute over those already in (Abbasi et al. 2022). Please explain or remove redundant results.
>
> Section 4.3 studies the effect of the degree of overlap (as an efficient mechanism for controlling the reusability and interference in the parameters) in representations and task similarity. The underlying framework (Active Dendrites + DANN) upon which the effect of heterogeneous dropout is studied is vastly different from the standard ANNs in Abbasi et al. 2022. We further show that in conjunction with the context-specific processing in Active Dendrites, heterogeneous dropout further facilitates the formation of task-specific subnetworks. This also sets up the stage for our subsequent analysis on the effect of dropout strength and task similarity. This is further studied in the new results on forward and backward transfer with different levels of task similarity and modularity (Section A.4).
>
> >Fig. 1(a): arrows are missing on the green line, and  seem to be referring to the same connection, $W_{ie}$ and $W_{ei}$  seem to be referring to the same connection
>
> Thank you for pointing this out. We have modified the diagram to make the connections more clearly and followed the same color scheme as in the main diagram. Hope this makes it more clear.
>
> Please see our general response for an overview of the changes in the revised manuscript. We hope to have addressed your concerns. Please let us know if we can provide further information or clarifications.
>
> [1] Tremblay, Robin, Soohyun Lee, and Bernardo Rudy. "GABAergic interneurons in the neocortex: from cellular properties to circuits." Neuron 91.2 (2016): 260-292

---

### Author Response · Authors · 2023-03-02
**Summary of changes in the revised manuscript**

We thank all the reviewers for their encouraging remarks and valuable feedback and suggestions. We tried to incorporate the suggested changes and additional experiments in the revised manuscript. Here we provide a summary of the changes (indicated with blue color in the manuscript)

- In Table 1, we added the upper (Joint) and lower (SGD) bounds. Added the results for Heterogeneous Dropout (HD) and ran all the experiments for 2 additional seeds. We now provide the mean and std of 5 runs.

- We conducted additional experiments on more challenging CL settings (Section 4.9). We added more results on Seq-MNIST and as a more challenging case, we evaluated our method on the grayscale version of the CIFAR10 dataset. We observe considerable improvements with the biological components.

- We extended the discussion (section 5) with additional limitations of our framework, particularly the use of backpropagation which is widely considered biologically implausible.

- We added a discussion on the training complexity of the different components of our framework (Section A.2)

- We conducted additional ablation (Section A.3) by adding the different biologically inspired components (activation sparsity, Heterogeneous Dropout, Synaptic Consolidation, and Experience Replay) on top of standard ANN instead of Active Dendrites + DANNs.

- We studied forward and backward transfer in the model (Section A.4) based on task similarity and the degree of modularity.

We hope that these additional experiments and changes combined with our individual responses to the specific questions of the reviewers address their main concerns. We would be happy to provide further information and engage in discussion.

---

### Decision · Action_Editors · 2023-03-24

**Recommendation:** Accept with minor revision

**Comment:**

The reviewers all appreciated that this was a comprehensive comparison of the role and interactions between a diverse set of bio-inspired innovations to ANNs that may improve continual learning.  They believed this is of value for the community.

Based on the initial paper, all reviewers wanted to see how the various innovations fared independently (in the absence of active dendrites).  This was added in what is now Table 6 (in appendix).
- I would recommend pointing to this table from the caption to Table 1.

Reviewers baVv and kt2N made a series of suggestions for changes and additional analyses, most of which, the authors have performed.

Reviewer kt2N was mostly happy with the changes, but made a few final requests that were accidentally commented without the authors' visibility.  I am copying these requests here and would encourage the authors to make updates reflecting these.
- "Could you explicitly write down the definitions of forward/backward transfer used in Tables 6 and 7? It's not 100% clear to me what you are measuring."
- "The result that active dendrites actually harm performance on Rotated-MNIST are quite surprising. This is not necessarily a result that disqualifies this work, but I think this result should be clearly reflected in the text. For example, in Section 4.1 the authors say: "Table 1 shows that models with feedforward neurons adhering to Dale’s principle perform as well as the standard neurons and can also further mitigate forgetting in some cases". This is not true for Rotated-MNIST. Similar corrections are needed in other sections."

There was a bit of discussion between authors and reviewers (especially reviewer baVv) about whether specific innovations are significant. From looking at table 1, which does not have indications of statistically significant improvements, I'm honestly not clear whether the Hebbian update is actually reliably improving performance by a significant amount.  In one place the authors added "Table 1 shows that this can lead to improvement in results."  However, since there are multiple comparisons, some of which go up and some go down, and with small effect sizes, it would be appropriate to perform significance testing, tone down the claims, and/or be clear that the effect size is quite small.

In addition to the points made by reviewers, I had a few minor comments about the presentation. A few terms are not introduced before their acronyms are used or before they are used in captions.
- "Heterogeneous dropout" is repeatedly mentioned in captions, but not fully introduced until the evaluation setting section.
- Table 1 would benefit from including all acronym definitions in the caption.  Many haven't been defined yet.


**Audience:**

The findings will be of interest to a subset of the TMLR audience.

**Claims And Evidence:**

This research work explores how combinations of previously proposed bio-inspired ANN innovations can support continual learning.  The claims are thoroughly tested and sufficiently convincing.

---

> ### Author Response · Authors · 2023-04-05
> **Authors response**
>
> We thank all the reviewers and Action Editors for their valuable feedback and suggestions. We are really glad that they unanimously believe that our study would be valuable to the research community. We hope our study will reach the right audience and inspire further work in this promising direction.
>
> We tried to incorporate all the suggestions to improve our manuscript. In the camera-ready version, we made the following changes:
> - Defined Heterogeneous Dropout in the contributions.
> - Added reference to Table 6 in Table 1 caption.
> - Added the descriptions of acronyms in Table 1.
> - Added the definitions of Forward Transfer and Forgetting in Section A.4.
> - Qualified the claims in Section 4.1, Section 4.3, and the Discussion section (in the discussion of the quality of context signal for Permuted-MNIST and Rot-MNIST, we also added a reference to the performance degradation in Rot-MNIST over standard ANN).
>
> Please let us know if we failed to make any necessary changes, we would be happy to revise the manuscript and incorporate any suggestions.